# Effect of rest, post-rest transport duration, and conditioning on performance, behavioural, and physiological welfare indicators of beef calves

Daniela M. Meléndez[1], Sonia Marti[2], Derek B. Haley[3], Timothy D. Schwinghamer[1], Xiaohui Yang[1], Karen S. Schwartzkopf-Genswein[1]*

**1** Lethbridge Research and Development Centre, Agriculture and Agri-Food Canada, Lethbridge, Alberta, Canada, **2** Department of Ruminant Production, IRTA, Caldes de Montbui, Barcelona, Spain, **3** Department of Population Medicine, University of Guelph, Guelph, Ontario, Canada

* karen.genswein@agr.gc.ca

## Abstract

The aim of this study was to assess the effects of conditioning, rest, and post-rest transport duration on welfare indicators of 6–7 mo old beef calves following a 20-h transport. Three hundred and twenty-eight weaned calves (237 ± 29.7 kg of BW) were randomly assigned to a 2 × 2 × 2 nested factorial design: conditioning, conditioned (**C**) or non-conditioned (**N**); rest, 0 (**R0**) or 8 (**R8**) h, and post-rest transport, 4 (**T4**) or 15 (**T15**) h. Calves were sampled before (LO1) and after (UN1) the initial 20-h journey, before (LO2) and after (UN2) the additional 4 or 15-h journey, and at 1, 2, 3, 5, 14, and 28 d after UN2. Data was analyzed using the GLIMMIX procedure of SAS. Fixed effects included conditioning, transport, and time nested within rest period, while random effects included animal and pen. Greater shrink ($p <$ 0.01) was observed in C than N calves after the initial 20-h transport. During the first week after transportation, the mean ADG of N calves was greater than C calves ($p < 0.01$). From d 14 to d 28, however, the mean ADG of C calves was greater than N calves ($p < 0.01$). Flight speed, cortisol and L-lactate concentrations were greater ($p \leq 0.05$) in C than N calves between LO1 and d 5, while greater ($p \leq 0.02$) non-esterified fatty acids, creatine kinase, serum amyloid-A, and haptoglobin concentrations were observed in N than C calves between LO1 and d 3. The R8-T4 calves had greater ($p < 0.01$) ADG than R8-T15 calves between LO1 and d 5. The R0-T4 calves had greater L-lactate concentrations than R0-T15 and R8-T4 calves (both $p = 0.02$) on d 1. The R0 calves had greater ($p < 0.01$) ADG than R8 calves between 14 and 28 d. This study suggests that C calves are better fit for transport than N calves as evidenced by behavioural and physiological parameters. Fewer and inconsistent differences were observed for rest and post-rest transport treatments.

**Data Availability Statement:** All relevant data are within the paper and its Supporting information files.

**Funding:** This project was funded by the Beef Cattle Research Council (http://www.beefresearch.ca/) and Agriculture and Agri-Food Canada (http://www.agr.gc.ca/eng/home/?id=1395690825741) Sustainable Beef and Forage Cluster under the Canadian Agricultural Partnership AgriScience Program (ANH.06.17 AIP-CL01). The co-author Sonia Marti was partly supported by the CERCA program from Generalitat de Catalunya. The funders had no role in study design, data collection and analysis, decision to publish, or preparation of the manuscript.

**Competing interests:** The authors have declared that no competing interests exist.

# Introduction

In North America, transport is an essential part of the beef industry as animals can be transported between ranches, backgrounding yards, feedlots, and abattoirs. The high visibility of animal transportation, especially in urban areas, contributed to an increase in public concern for the welfare of animals during transportation [1]. This has led to amendments to Canada's *Health of Animals Regulations* in 2019 which included a reduction in the length of time cattle can be transported from 48 to 36 h for weaned cattle and to 12 h for unweaned calves; while the minimum length of rest en route was increased from 5 to 8 h [2]. However, there are few and contradictory studies assessing the effects of rest on cattle welfare during transportation.

Newly weaned calves that were rested for 0 or 5 h had improved welfare over calves rested for 10 or 15 h [3] and preconditioned calves benefited from a 2 h rest compared to calves that were not provided with any rest [4]. Contrary to the previous studies, no behavioural or physiological differences were observed between conditioned calves receiving a 0, 4, 8, or 12 h rest [5].

Within Canada, feedlots in Central Canada often acquire additional weaned beef calves from both Western and Atlantic Canada. Currently there are at least four commercial rest stop facilities in Ontario where cattle can be off-loaded and rested (two located near Thunder Bay, and two at Kapuskasing and Hallebourg) which is a mid-point for typical routes of cattle moving east to west or vice versa [6]. A sister transport study (data not published) reported that cattle in Canada travel an average of 22 h before they arrive at these rest stops and need to be transported, on average, an additional 16 h before they reach their final destination. Consequently, transport durations before and after a rest period could have similar impacts on calf welfare. In two previous studies assessing the effect of rest on the welfare of cattle under Canadian conditions, which were conducted by the authors, conditioning and longer transport had large effects, while the effects due to rest duration were minimal [5, 7].

To date, no studies have evaluated the effects of long versus short transport durations after a rest period. Therefore, the aim of this study was to assess the effect of conditioning, rest, and transport duration following the provision of a rest, on welfare indicators in 6-7-month-old beef calves transported by road.

# Materials and methods

This study was approved by the Animal Care Committee of Lethbridge Research and Development Centre (LeRDC) (ACC number 2011). Calves were cared for in accordance with the Canadian Council of Animal Care [8].

With the exception of the treatment groups and number of animals, the materials and methods for behavioural (standing and lying behaviour, calf attitude and gait score, feeding behaviour, and flight speed) and physiological parameters (weight, rectal temperature, and blood sampling) are the same as the two related studies conducted previously by the authors [5, 7].

## Animal management and transport

Three hundred and twenty-eight crossbred steer calves—Black or Red Angus × Hereford/Simmental and Black or Red Angus × Charolais (237 ± 29.7 kg of BW) were sourced from two different ranches in southern Alberta, Canada. Treatments consisted of a 2 × 2 × 2 factorial design where the main factors included conditioning: conditioned (**C**; $n$ = 164) or non-conditioned (**N**; $n$ = 164); rest: 0 h (**R0**; $n$ = 164) or 8 h (**R8**; $n$ = 164), and transport duration after rest: 4 h (**T4**; $n$ = 164) or 15 h (**T15**; $n$ = 164). Calves were divided into two groups (Group 1 and 2) and each group was transported by road for 20 h, rested, and transported for an

**Table 1. Chronology of sampling for Group 1 and 2 of conditioned and non-conditioned, crossbred beef calves, transported for 20 h, rested for 0 (R0) or 8 (R8) h, and transported for 4 (T4) or 15 (T15) h after rest.**

| Samples | Group 1 | | | | Group 2 | | | |
|---|---|---|---|---|---|---|---|---|
| LO1 | Nov 16th | | | | Nov 24th | | | |
| | 1329–1734 | | | | 1300–1716 | | | |
| UN1 | Nov 17th | | | | Nov 25th | | | |
| | 1415–1709 | | | | 1349–1743 | | | |
| | **R0** | | **R8** | | **R0** | | **R8** | |
| LO2 | - | | Nov 18th | | - | | Nov 26th | |
| | | | 0111–0223 | | | | 0102–0211 | |
| | **T4** | **T15** | **T4** | **T15** | **T4** | **T15** | **T4** | **T15** |
| UN2 | Nov 17th | Nov 18th | Nov 18th | Nov 18th | Nov 25th | Nov 26th | Nov 26th | Nov 26th |
| | 1956–2052 | 0651–0740 | 0744–0824 | 1734–1817 | 1910–2005 | 0637–0724 | 0733–0817 | 1736–1827 |
| 1 d | Nov 18th | Nov 19th | Nov 19th | Nov 19th | Nov 26th | Nov 27th | Nov 27th | Nov 27th |
| | 1945–2039 | 0642–0726 | 0727–0805 | 1731–1812 | 1916–2005 | 0629–0709 | 0710–0754 | 1929–2013 |
| 2 d | Nov 19th | Nov 20th | Nov 20th | Nov 20th | Nov 27th | Nov 28th | Nov 28th | Nov 28th |
| | 1940–2018 | 0633–0706 | 0707–0742 | 1725–1803 | 1909–1949 | 0625–0658 | 0700–0732 | 1723–1758 |
| 3 d | Nov 20th | Nov 21st | Nov 21st | Nov 21st | Nov 28th | Nov 29th | Nov 29th | Nov 29th |
| | 1944–2023 | 0631–0716 | 0718–0753 | 1731–1816 | 1912–1951 | 0626–0703 | 0704–0749 | 1728–1808 |
| 5 d | Nov 23rd | | | | Dec 1st | | | |
| | 0757–1104 | | | | 0802–1047 | | | |
| 14 d | Dec 2nd | | | | Dec 10th | | | |
| | 0802–1052 | | | | 0802–1046 | | | |
| 28 d | Dec 16th | | | | Dec 24th | | | |
| | 0804–1043 | | | | 0757–1101 | | | |

Values indicate the date and time (24 h clock) sampling took place.

Calves were sampled before (LO1) and after (UN1) a 20-h transport, provided with either no rest (R0) or 8 h (R8) of rest, and sampled before (LO2) and after (UN2) an additional 4 (T4) or 15 (T15) h transport, as well as on d 1, 2, 3, 5, 14, and 28 after UN2.

additional 4 or 15 h (Table 1). The two groups were transported 8-d apart. Samples were collected prior to loading (LO1) and after unloading (UN1) following the 20-h transport, as well as prior to loading (LO2) and after unloading (UN2) following the additional 4 or 15-h transport (Fig 1). Calves were also sampled on d 1, 2, 3, 5, 14, and 28 after UN2. To avoid variation in physiological parameters due to the circadian rhythm, calves were sampled 24 (1 d), 48 (2 d), and 72 (3 d) h after UN2. Calves were randomly assigned to treatments (40 calves/treatment) and pens (10 calves/pen). Due to differences in rest and transport duration, UN2 sampling for the R0-T4 group was 10, 11, and 23 hours prior to UN2 sampling for R8-T4, R0-T15, and R8-T15 calves, respectively (Table 1). Therefore d 5, 14, and 28 are the equivalent of d 6, 15, and 29 for R0-T4 calves. During the study calves from different treatments were kept in separate pens.

**Conditioned calves.** Twenty-eight d prior to LO1 (October 19th and 27th, 2020), two groups of eighty-two calves were weaned and transported for approximately 1 h from the

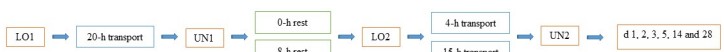

**Fig 1. Timeline of calves were sampled before (LO1) and after (UN1) a 20-h transport, provided with either no rest (R0) or 8 h (R8) of rest, and sampled before (LO2) and after (UN2) an additional 4 (T4) or 15 (T15) h transport, as well as on d 1, 2, 3, 5, 14, and 28 after UN2.**

ranch of origin to the LeRDC. Calves were processed the day after arrival, which included receiving a 7-way bovine clostridial vaccine (Ultrabac/Somubac, Zoetis Canada Inc., Kirkland, QC, Canada); a 5-way bovine viral diarrhea, rhinotracheitis, parainfluenza, and bovine respiratory syncytial virus vaccine (Pyramid FP 5 + Presponse SQ, Boehringer Ingelheim., Burlington, ON, Canada); an antibiotic (Draxxin, Zoetis Canada Inc., Kirkland, QC, Canada); an anti-parasitic agent (Ivomec Pour-on for Cattle, Boehringer Ingelheim, Burlington, ON, Canada); and an ear tag and a half duplex RFID tag. During the conditioning period (28 d) calves were housed in 4 pens (36.7 m × 22.2 m) with a central water trough, with 41 animals per pen. During the first 5 d after arrival to the LeRDC feedlot calves received an *ad libitum* diet consisting of 65% corn silage, 20% alfalfa hay, 13% barley grain, and 2% supplement with vitamins and minerals. On d 6, calves received an *ad libitum* diet consisting of 75% corn silage, 10% alfalfa hay, 13% barley grain, and 2% supplement with vitamins and minerals. From d 7 to 28, calves received *ad libitum* feed consisting of 85% corn silage, 13% barley grain, and 2% supplement with vitamins and minerals.

Calves in the present study will be referred to as 'conditioned' calves, because the suggested preconditioning time period prior to shipping of 30 to 45 d was not met [9, 10]. A conditioning period of 20 d was selected to match the methods of two previous related transport studies that reported inconsistent differences in welfare indicators of calves that received 0 or 8 h of rest [5, 7]. However, a snow storm delayed the trial (for both groups) by one week, and therefore increased the length of preconditioning to 28 instead of 20 d.

**Non-conditioned calves.** Prior to LO1 (November 16th and 24th, 2020), two groups of eighty-two calves were separated from their dams and transported for approximately 1 h from the ranch of origin to the LeRDC. Non-conditioned calves received an ear tag and an RFID tag during LO1 sampling for identification. Processing (vaccine, antibiotic, and anti-parasitic administration) was postponed until UN2 to simulate industry practices, where calves are typically processed on arrival to the feedlot.

**Housing and feeding.** After the initial 20-h transport, calves (10/pen) were housed in 32 pens (21 × 27 m) with a fence line water trough. Non-conditioned calves received the same transitioning diets that the conditioned calves received during the conditioning period as previously described. During the first 5 d after arrival to the LeRDC, non-conditioned calves received an *ad libitum* diet consisting of 65% barley silage, 20% alfalfa hay, 13% barley grain, and 2% supplement with vitamins and minerals. From d 6 to d 20, non-conditioned calves received an *ad libitum* diet consisting of 75% barley silage, 10% alfalfa hay, 13% barley grain, and 2% supplement with vitamins and minerals. From d 20 to 28, non-conditioned calves received 85% corn silage, 13% barley grain and 2% supplement with vitamins and minerals.

After the 20 h transport, conditioned calves received *ad libitum* feed consisting of 85% corn silage, 13% barley grain and 2% supplement with vitamins and minerals to meet beef cattle nutrition requirements [11].

**Transport.** Three Merritt 53' quad-axle trailers bedded with wood shavings were used to transport the calves for 20 h as well as for the additional 4 and 15 h after rest. During the 20 h transport, calves were placed in the nose (*n* = 10), deck (*n* = 27), belly (*n* = 27), back (*n* = 13), and doghouse (*n* = 5) compartments (Fig 2). Loading densities were: nose 0.77, deck 0.68, belly 0.68, back 1.10, and doghouse 2.22 m²/animal. Treatments were equally distributed by compartments. During the additional 4 and 15 h transport, calves were placed in the nose (*n* = 10; 0.77 m²/animal) and belly (*n* = 30–32; 0.57 to 0.61 m²/animal). For the initial 20-h transport, trailers travelled together to ensure similar road and environmental conditions. For the additional 4 and 15-h transport, trailers followed the same route to ensure similar road conditions. Drivers monitored the calves when stopped for rest and ensured that calves were standing to avoid injuries.

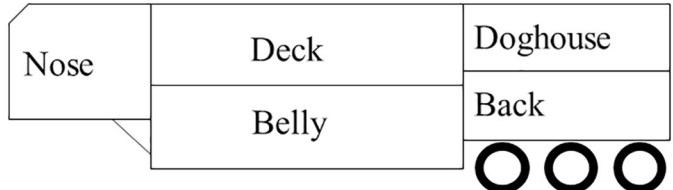

**Fig 2. Layout of five compartments within a quad-axle livestock trailer.**

Air temperature and humidity within the trailers were recorded using the DS1923 hygrochron temperature/humidity logger iButton (Maxim Integrated Products, Sunnyvale, CA, USA). Loggers were zip tied to the front of the ear tags of a subset of 128 calves, which were equally distributed by compartment. Relative humidity (RH, %) and temperature (T, ˚C) data were collected every 5 minutes during the 20, 4, and 15 h transport and were used to calculate a temperature humidity index (THI) (Table 2) using the following formula:

$$THI = 0.8 \times T + RH \times (T - 14.4) + 46.4$$

Stress categories for THI have been previously defined as alert (75 to 78), danger (79–83) or emergency (84 or higher) [12].

### Behavioural assessments

**Standing and lying.**   Behaviour was assessed as previously described by Meléndez et al. [5, 7]. Standing and lying behaviour of a subset of 12 calves/treatment was recorded with accelerometers (Hobo pendant G, Onset Computer Corporation, Bourne, MA, USA) attached to the right hind leg of the calves using Vet Wrap (Professional Preference, Calgary, AB, Canada). At LO1, accelerometers were placed, each in a vertical position on the right hind leg of the calf with the X-axis pointing up towards the backbone of the calf, and set to record data at 1-min intervals. Data from the days when the accelerometers were placed (d -1) and removed (d 6) from the calves were excluded from the analysis due to data not being collected for a full 24 h period. Daily standing and lying percentage, and daily standing and lying mean bout duration were summarized for further analysis for 5 d after transport. Data regarding the duration of time spent standing during transportation (20, 4, and 15 h) and rest (8 h in the pen) were extracted from the accelerometer raw data and summarized by hour for further analysis.

**Table 2. Temperature and humidity index (THI) within trailers used to transport two groups of 6-7-mo-old beef calves (Group 1 and 2) for 20, 4, and 15 h transport.**

| | Temperature Humidity Index (THI) | | |
|---|---|---|---|
| | Minimum | Maximum | Average |
| *Group 1* | | | |
| 20h | 14.4 | 88.6 | 41.2 |
| 4h | 35.4 | 77.4 | 52.4 |
| 15h | 26.0 | 81.9 | 50.0 |
| *Group 2* | | | |
| 20h | 31.4 | 84.0 | 47.0 |
| 4h | 31.4 | 80.6 | 45.4 |
| 15h | 30.9 | 88.2 | 46.6 |

**Calf attitude and gait score.**   Calf attitude and gait were assessed after UN1 and UN2 as described previously by Meléndez et al. [5, 7]. An experienced observer (KSSW) assessed calves after exiting the squeeze chute while walking down an alley outside of the handling facilities.

Attitude was evaluated using a 4-point scale [13]. Normal, bright, and alert cattle that held their head up and readily moved away from the observer received a score of 0. Cattle that were slightly depressed but responded quickly to the observer and appeared normal received a score of 1. Cattle that were moderately depressed, stood with their head down, ears drooped, had an abdomen that lacked fill and appeared floppy, and moved away slowly from the observer received a score of 2. Cattle with severe depression, that stood with their head down, were very reluctant to move, and had very noticeable gauntness of abdomen received a score of 3.

Gait score was evaluated using a 5-point scale [14]. Cattle that walked normally, with no apparent lameness or change in gait were characterized as "walking normally" and received a score of 0. Cattle that walked easily and readily, the line of the backbone was normal and they were able to bear full weight on all four limbs, but had an observable gait alteration were characterized as "mildly lame" and received a score of 1. Cattle that were reluctant to walk, did so with a short weight-bearing phase of stride, rested the affected limb when standing, and exhibited increased periods of recumbency were characterized as "moderately lame" and received a score of 2. Cattle that laid down most of time, were reluctant to stand, refused to walk without stimulus, did not bear weight on the affected limb and "hopped" when moving, did not use all limbs when standing and had an arched backbone with a caudoventral tip to the pelvis were characterized as "severely lame" and received a score of 3. Cattle that were recumbent, unable to rise, and requiring euthanasia were characterized as "non-ambulatory" and received a score of 4.

**Dry matter intake and feeding behaviour.**   Dry matter intake (DMI; kg/d/h) was determined as previously described by Meléndez et al. [5, 7] by pen feed refusals recorded daily for d 0, 1, and 2 in relation to UN2, and weekly (week 1, 2, 3, and 4) until d 28 after transport for 24 pens. Feed samples were collected on feed refusal days to determine feed dry matter intake (DMI).

Individual feeding behaviour was monitored in the remaining 8 pens using the GrowSafe feed bunk monitoring system (GrowSafe® Systems, Airdrie, AB, Canada) as previously described by Meléndez et al. [5, 7]. Calves were fitted with radio frequency ear tags (RFID, All-flex Livestock Intelligence, St-Hyacinthe, QC, Canada) and each pen was equipped with two tubs that recorded individual feed intake during the study period. Feeding data was used to calculate meal size (kg/meal/day), meal duration (min/meal/day), meal frequency (meal/day), feeding intake (kg/day), feeding duration (min/day) and feeding rate (g/min/day). A meal criterion of 300 s was selected based on previous studies in cattle [15, 16]. Feeding behaviour was evaluated for 1 pen per treatment (10 animals per treatment) for group 1. One calf from each of the treatment groups N-R0-T15, N-R0-T4, and N-R8-T15 housed in pens 6, 7, and 8 did not adapt to feeding from the GrowSafe tubs and therefore were moved to a pen with regular feed bunks. Therefore, 9 animals instead of 10 were housed in pens 6, 7, and 8.

**Flight speed.**   The velocity at which animals exited the chute was collected as previously described by Meléndez et al. [5, 7] at LO1, UN1, LO2, UN2, 1, 2, 3, 5, 14, and 28 d. The time it took an animal to travel 2 m was electronically recorded using two sets of light beams as described previously by Burrow et al. [17].

## Physiological assessments

**Weight and rectal temperature.**   Calves were weighed while standing in a hydraulic squeeze chute (Cattlelac Cattle, Reg Cox Feedmixers Ltd, Lethbridge, AB, Canada) equipped

with a weigh scale and rectal temperature was measured using a Sharptemp V digital ther-mometer (Cotran Corporation, Portsmouth, RI, USA) at each sampling point (LO1, UN1, LO2, UN2, 1, 2, 3, 5, 14, and 28 d after UN2).

Average daily gain (ADG) was calculated for the first week after transport by subtracting d 5 body weight (BW) from the initial (LO1) BW and dividing by the number of days on trial (6 d). The ADG of the second week was calculated by subtracting the d 14 BW from the d 5 BW and dividing by the number of days between sampling points (9 d). The ADG of the third and fourth week was calculated by subtracting d 28 BW from d 14 BW and dividing by the number of days between samples (14 d). Percentage shrink for the 20 h-transport (shrink 1) was calcu-lated using BW collected at LO1 and UN1. Percentage shrink for the additional 4 and 15 h-transport (shrink 2) was calculated using the BW collected at UN1/LO2 and UN2. The BW collected at UN1 was used for R0 calves because calves did not receive a rest, while BW col-lected at LO2 was used for R8 calves. Shrink was calculated using the formula: shrink = (1 − (BW after transport / BW before transport) ×100).

**Blood samples.** Blood samples were collected as previously described by Meléndez et al. [5, 7]. Blood samples were collected from a subset of 12 calves/treatment using jugular veni-puncture at LO1, UN1, LO2, UN2, 1, 2, 3, 5, 14, and 28 d after UN2. Blood samples were col-lected into 3, 10-mL non-additive tubes (BD vacutainer; Becton Dickinson Co., Franklin Lakes, NJ, USA) for cortisol, non-esterified fatty acids (NEFA), haptoglobin (HP), serum amy-loid-A (SAA), and creatine kinase (CK) analysis. Blood samples were also collected into a 7-mL sodium fluoride tube (BD vacutainer; Becton Dickinson Co., Franklin Lakes, NJ, USA) for L-lactate analysis, and into a 6-mL EDTA tube (BD vacutainer; Becton Dickinson Co., Franklin Lakes, NJ, USA) to determine complete blood cell count (CBC). Samples collected into the non-additive tubes and the sodium fluoride tube were left at room temperature for 1 h prior to centrifugation for 15 min at $2.5 \times g$ at 4˚C. Serum was decanted and frozen at -20˚C for further analysis.

NEFA were analyzed as an indicator of fat mobilization due to feed deprivation. NEFA con-centrations were quantified using a colorimetric assay (HR Series NEFA-HR (2), FUJIFILM Wako Pure Chemical Corporation, Osaka, Japan). The intra-assay CV was 4.1% and the inter-assay CV was 14.0%. L-lactate was measured as an indicator of muscle damage using an L-lac-tate colorimetric assay (Lactate Assay Kit, Cell Biolabs, Inc., San Diego, CA, USA) to quantify L-lactate concentrations in serum. The intra-assay CV was 4.2% and the inter-assay CV was 14.3%. CK concentrations were quantified as an indicator of muscle damage using a colorimet-ric assay (EnzyChrom™ Creatine Kinase Assay Kit, BioAssay Systems, Hayward, CA, USA). The intra-assay CV was 4.4% and the inter-assay CV was 9.9%. HP and SAA were analyzed as indicators of stress, inflammation, infection and trauma. HP concentrations were quantified using a colorimetric assay (Tridelta Development Ltd., Maynooth, Co, Kildare, Ireland), while SAA concentrations were quantified using an enzyme linked immunosorbent assay (Tridelta Development Ltd., Maynooth, Co, Kildare, Ireland). The HP intra-assay CV was 6.3% and the inter-assay CV was 5.4%. The SAA intra-assay CV was 5.6% and the inter-assay CV was 75.0%. Data for SAA was limited to LO3, UN2, d 1, 2, and 3 sampling points due to the high number of samples that had to be diluted due to high SAA concentrations. Samples were either diluted to a 1:500, 1:1000, 1:2000 or 1:4000 ratio. Complete blood cell count (CBC) was mea-sured as an indicator of immune function using a HemaTrueHematology Analyzer (Heska, Loveland, Co). Serum cortisol concentrations were collected as an indicator of acute stress and concentrations were quantified using an immunoassay kit (DetectX Kit, Arbor Assays, Ann Arbor, MI, USA). The intra-assay CV was 7.8% and the inter-assay CV was 13.8%.

**Morbidity and mortality.** Morbidity and mortality of experimental animals were recorded over a 28-d experimental period.

## Sample collection

Weight and rectal temperature were collected from the experimental calves ($n$ = 328) and a subset of 12 calves/treatment (3 calves/pen) were sampled for physiological and behavioural indicators of welfare. Calves were sampled prior to (LO1) and after (UN1) the 20 h transport, and prior to (LO2) and after (UN2) the additional 4 and 15 h transport. In addition, calves were sampled on d 1, 2, 3, 5, 14 and 28 after UN2.

For LO1, calves were sampled in groups of 20, alternating between conditioned and non-conditioned calves. After LO1 and LO2 sampling, calves were sorted into 5 pens, in order for treatments to be equally distributed into one of the 5 compartments within the trailer.

**Statistical analysis.** Data were analyzed using mixed models due to the inclusion of fixed effects: conditioning, time (nested in rest), and transport duration after rest and random effects: animal and pen. Time was nested in rest to account for the missing sampling point (LO2) for the R0 treatment calves, which did not receive a rest. Data were analyzed using PROC GLIMMIX (SAS, version 9.4, SAS Inst. Inc., Cary, NC). Distributions from the exponential family (gamma, inverse Gaussian, log-normal, normal, exponential, and shifted $t$) for each variable were tested and selected based on the Bayesian information criterion (BIC). After selecting the distribution, covariance structures: compound symmetry (CS), heterogeneous compound symmetry (CSH), variance component structure (VC), first-order autoregressive (AR1), and heterogeneous first-order autoregressive (ARH1) were tested and selected based on the BIC value (S1 Table).

Covariates in the model varied depending upon the variable assessed. Group, time of day, breed, and flight speed were included as covariates in the analysis of NEFA, L-lactate, HP, SAA, cortisol, CK, and CBC. Group and breed were included as covariates for the analysis of BW, ADG, rectal temperature, shrink 1 and shrink 2. Breed was included as a covariate for the analysis of GrowSafe data and group was included as a covariate for the analysis of DMI. Breed and group were included as a covariate for the analysis of accelerometer data. GrowSafe and accelerometer data collected on d 0 were adjusted to the proportion of time animals were in the pen, as this varied between treatments. Results are reported as least squares-means ($\mu$) including the upper (u) and lower (l) limits at 95% confidence. SAS PROC GLIMMIX iterated 1000 times at multiple levels of iterations (MAXOPT = 1000; NLOPTIONS MAXITER = 1000). Bonferroni's correction for multiple comparisons was used.

Statistical significance was $p \leq 0.05$. In some cases, the $F$-test indicated the statistical significance of an interaction but there were no differences between specific comparisons of interest and therefore those interactions are not discussed. Statistically significant differences that where reported were limited to comparisons of interest: comparisons between treatments with the same conditioning and rest effect, but differing transport effect (e.g. C-R0-T4 vs C-R0-T15); the same rest and transport effect, but differing conditioning effect (e.g. C-R0-T4 vs N-R0-T4); or the same conditioning and transport effect, but differing rest effect (e.g. C-R0-T15 vs C-R8-T15) at the experimental sampling points.

Data from 2 accelerometers (both from the C-R0-T15 treatment group) were excluded from the analysis due to the accelerometers being lost during transportation. Lying and standing bout duration was limited to d 2 to 5 as there was no bout duration for d 1. Attitude and lameness scores of 2 animals were not recorded because their observation was missed.

## Results and discussion

### Standing and lying

Treatments key: conditioned (C) and non-conditioned (N) calves, rested for 0 (R0) or 8 (R8) h and transported for and additional 4 (T4) or 15 (T15) h of transport after rest.

**Hourly standing percentage during the initial 20 h of transport.** A three-way interaction for transport × conditioning × time (nested in rest) effect ($p$ = 0.01) was observed for mean standing percentage. Differences in standing percentage were observed during the initial 20 h transport at hours 4, 10, and 11. The N-R0-T15 and C-R8-T15 calves had greater ($p$ <0.01) standing percentage than N-R8-T15 calves at 4 h of transport (Fig 3A). The C-R0-T4, N-R8-T4, and N-R0-T15 calves had greater ($p$ <0.01) standing percentage than N-R0-T4 calves at 10 h of transport. The N-R8-T4 calves had greater ($p$ <0.01) standing percentage than N-R0-T4 calves, while N-R8-T15 calves had greater ($p$ <0.01) standing percentage than N-R8-T4 calves at hour 11 of transport. The results observed at 4 and 10 h are expected, as N calves may be more exhausted due to weaning and transportation just prior to the 20 h transport. However, we would have expected to see differences between all treatments that had C calves compared to all treatments that had N calves, which was not the case. These results are contrary to results observed in a previous study where N calves had greater standing percentage than C calves at hours 30 and 32 of transport of a 36 h journey. In the present study, differences were observed in the first quarter and first half of the journey while in the previous study differences where observed towards the last quarter of the journey. Differences between studies could be due to differences in treatments (conditioning, source, and rest vs conditioning, rest, and post-rest transport duration).

Previously described differences between rest (R0 vs R8) and post-rest transport duration (T4 vs T15) treatments were unexpected during the initial 20 h-transport since calves had not yet been exposed to their assigned rest (0 or 8 h) and transport duration (4 or 15 h) after rest treatments. Space allowance had no impact on the treatment differences observed because animals from each treatment group were equally distributed amongst the compartments.

**Hourly standing percentage during the 8h rest period.** A transport × conditioning × time (nested in rest) effect ($p$ = 0.02) was observed for mean standing percentage, where C-T15 calves had greater ($p$ < 0.01) standing percentage than C-T4 calves during the first hour of rest, while C-T4 calves had greater ($p$ < 0.01) standing percentage than C-T15 calves at hour 8 of the rest period (Fig 3B). These findings are difficult to explain because calves were transported and rested similarly (i.e. they had not yet gone on their post-rest transport). Although rest periods are known to vary throughout the day [18] this was not a factor in this study because calves were provided with a rest during the same hours of the day. In addition, pen conditions did not vary between treatment groups as all calves were provided with a rest on the same day and at the same time. Differences observed during the 8 h rest could be due to individual animal differences in time budgets [18, 19].

**Hourly standing percentage during the additional 4 h transport.** A time (nested in rest) effect ($p$ < 0.01) was observed for mean standing percentage. The R0 $\mu$ = 99% (u = 143.4, l = 69.2) calves had greater ($p$ < 0.01) standing percentage than R8 $\mu$ = 38% (u = 55.3, l = 26.2) calves during the first hour of transport (Fig 3C). No differences were observed between hours 2 and 4 of transport but R0 calves had numerically (82 to 92%) greater standing percentage than the R8 group (51 to 63%). These results are contrary to our previous study, where R8 calves had greater standing percentage at 3 h of transport and numerically greater standing percentage during the entire 4 h transport compared to R0 calves [7]. Differences in standing percentage between the present and previous study [7] could be due to differences in the time of day when R0 (0700–1100 vs 1500–1900) and R8 (1700–2100 vs 0100–0500) calves were transported for an additional 4 h. Unrested calves (R0) in the current study should have been more fatigued than rested (R8) counterparts and therefore we hypothesized would have reduced standing percentages during the 4 h transport compared to the R8 group. Differences observed between R0 and R8 calves could be due to the time of day and THI in which the additional 4 h transport took place in the present study, as R0 calves were transported at

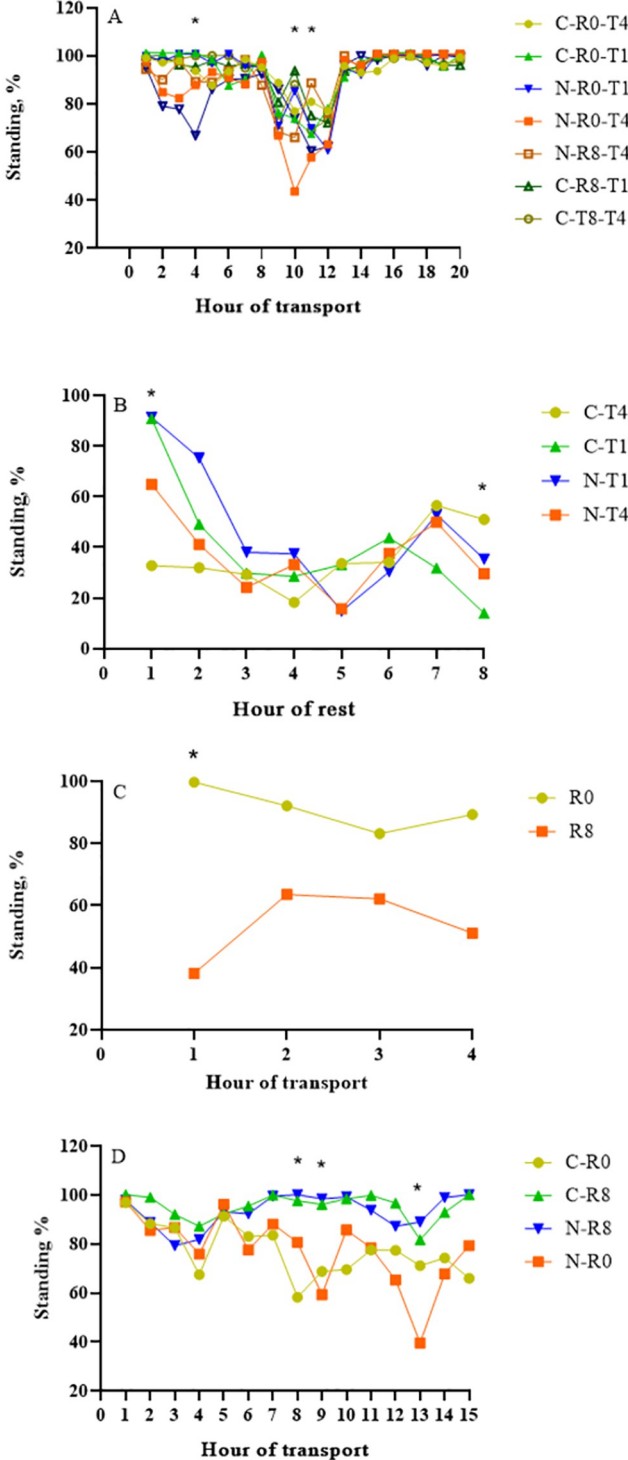

**Fig 3. Least squares-means (± upper and lower limits at 95% confidence) of standing percentage (%) during the (A) 20 h transport, (B) 8 h rest period, (C) additional 4 h and (D) 15 h transport of conditioned (C) and non-conditioned (N), calves rested for 0 (R0) or 8 (R8) h and transported for an additional 4 (T4) or 15 (T15) h.** *Indicates significant differences between treatments ($p < 0.05$).

approximately 1500, while R8 calves were transported at 0200. Although we expected calves to stand during transportation, it is possible that standing percentage of calves in the R0 group was greater than calves in the R8 group during the first hour of transport because R8 calves were transported during a period of time when calves would typically rest [18]. Previous studies assessing standing and lying behaviour during transport reported that the majority of calves stand during road transportation and lay down towards the end of the journey [20, 21]. However, we were not able to compare our results to previous studies assessing the effect of rest, because behaviour during transportation in the previous studies was not assessed [3, 4].

**Hourly standing percentage during the additional 15 h transport.** A conditioning × time (nested in rest) effect ($p = 0.04$) was observed for mean standing percentage during the 15 h transport. Differences were observed between treatments at hour 8, 9, and 13 of transport. The C-R8 calves had greater ($p < 0.01$) standing percentage than C-R0 calves at 8 h of transport (Fig 3D), while the N-R8 calves had greater ($p = 0.01$) standing percentage than N-R0 calves at 9 h of transport. The C-R0 calves had greater ($p < 0.01$) standing percentage than N-R0 calves, while N-R8 calves had greater ($p = 0.01$) standing percentage than N-R0 calves at hour 13 of transport. These findings were expected because conditioning has been shown to improve calf fitness for transport [5, 7]. In addition, rested calves (R8) would be expected to stand more than those not receiving a rest (R0). Different standing percentage results were observed during the additional 4 and 15 h of transport. Overall, R0 calves had greater standing percentage in the 4 h transport, while R8 calves had greater standing percentage during the 15 h transport. Although time of day could have influenced standing/lying behaviour, both the 4 and 15 h trailers departed at the same time of day (around 1500 for R0 and 0200 for R8). The difference in number of sampling points between 4 and 15 h transport, which was summarized by hour, could have influenced the results observed between treatments as a repeated measure analysis was used.

**Daily standing and lying during 1 to 5 d after transport.** A conditioning × time (nested in rest) effect ($p = 0.01$) was observed for mean standing percentage on d 1, where C-R0 $\mu = 26\%$ (u = 28.9, l = 23.9) and N-R0 $\mu = 31\%$ (u = 34.5, l = 28.7) calves had greater ($p < 0.01$) standing percentage than C-R8 $\mu = 21\%$ (u = 22.8, l = 19.0) and N-R8 $\mu = 20\%$ (u = 22.5, l = 18.8) calves, respectively. These results are contrary to what we expected as R0 calves would likely be more fatigued on d 1 after transport and would therefore stand less than R8 calves. Due to differences in rest and transport duration, R0-T4, R0-T15, R8-T4, and R8-T15 calves spent 24, 16, 16, and 6 h in the pen on d 1, respectively. Both R8-T4 and R0-T15 calves returned to the pen at a similar time (between 0724 and 0830) while R8-T15 calves returned to the pen at a later time (1817 to 1827) of the day. Differences observed between R0 and R8 calves could be due to differences in the time that calves arrived to the pen (morning vs afternoon) which has an effect on behaviour [18]. In addition, differences observed could be due to adjusting standing percentage on d 1, which was done by adjusting standing percentage observed in the pen to represent what standing percentage would have been if calves were for 24 h in the pen. This adjustment was done with the intention to make a fair comparison between treatments, however, it is possible that when making this adjustment we are erroneously extrapolating the behaviour observed after transport to the rest of the day. This adjustment may not have been an issue for R0-T4, R0-15, and R8-T4 calves which were assessed for 24 and 16 hours, but it could potentially affect the standing percentage of R8-T15 calves which were assessed for 6 h in the evening after a 15 h transport. Although differences were only observed between R0 and R8 calves, post-transport duration (T4 and T15) was mentioned in addition to rest as these were the factors that determined the time of day animals returned to their home pens and the length of time they spent in the pen for d 1.

A transport × time (nested in rest) effect ($p = 0.01$) was observed for mean standing percentage, where on d 1, R0-T4 $\mu = 28\%$ (u = 31.1, l = 25.9), R0-T15 $\mu = 29\%$ (u = 32.0, l = 26.5), and R8-T4 $\mu = 23\%$ (u = 25.4, l = 21.2) calves had greater ($p \leq 0.02$) standing percentage than R8-T4 $\mu = 23\%$ (u = 25.4, l = 21.2), R8-T15 $\mu = 18\%$ (u = 20.2, l = 16.8), and R8-T15 $\mu = 18\%$ (u = 20.2, l = 16.8) calves, respectively. The R8-T4 $\mu = 30\%$ (u = 33.2, l = 27.7) calves had greater ($p = 0.05$) standing percentage than R8-T15 $\mu = 25\%$ (u = 27.6, l = 23.1) calves on d 2. These findings were opposite to the results observed for the additional 15 h of transport, where the R8 calves had greater standing percentage than the R0 calves. It is possible that during the first day after transport, R0 calves stood more because they rested more during the 15 hours of transport (previous results) and therefore spent more time standing on d 1. The T4 calves had greater standing percentage than T15 calves on d 1 and 2. Differences observed between treatments could be due to differences in transport duration as we would expect calves transported for an additional 11 h to experience more fatigue [5], or due to adjusting standing percentage on d 1 based on the hours that calves were in the pen as mentioned in the previous paragraph.

A conditioning × time (nested in rest) effect ($p = 0.03$) was observed for mean standing bout duration, where N-R0 $\mu = 59$ min (u = 69.5, l = 51.7) calves had greater ($p = 0.01$) standing bout duration than C-R0 $\mu = 42$ min (u = 49.1, l = 36.3) calves, while N-R8 $\mu = 52$ min (u = 60.5, l = 45.4) calves had greater ($p < 0.01$) standing bout duration than C-R8 $\mu = 34$ min (u = 39.4, l = 29.6) calves on d 2. Differences between treatments could be due to N calves being more restless than C calves due to weaning, as newly weaned calves have been reported to walk more in an attempt to reunite with the cow [22].

A transport × time (nested in rest) effect ($p = 0.05$) was observed for mean standing bout duration, where R0-T15 $\mu = 51$ min (u = 59.8, l = 43.9) calves had greater ($p = 0.02$) standing bout duration than R8-T15 $\mu = 36$ min (u = 42.2, l = 31.6) calves, while R8-T4 $\mu = 48$ min (u = 56.5, l = 42.4) calves had greater ($p = 0.05$) standing bout duration than R8-T15 $\mu = 36$ min (u = 42.2, l = 31.6) calves on d 2. Differences observed for greater standing bout durations were unexpected, as R0 calves did not receive a rest and therefore may have been more fatigued, standing for shorter periods of time compared to R8 calves that had been rested. Differences observed for standing bout duration for T4 calves were expected as animals were transported for a shorter period of time than T15 calves and therefore would be less fatigued which helps to explain the fact that they stood more on d 2 than T15 calves.

A conditioning × time (nested in rest) effect ($p = 0.01$) was observed for mean lying bout duration, where N-R0 $\mu = 88$ min (u = 102.9, l = 75.6) calves had greater ($p = 0.04$) lying bout duration than C-R0 $\mu = 63$ min (u = 74.7, l = 54.5) calves, while N-R8 $\mu = 99$ min (u = 115.2 = 85.3) calves had greater ($p < 0.01$) lying bout duration than C-R8 $\mu = 66$ min (u = 77.9, l = 57.6) calves on d 2. The N-R8 $\mu = 90$ and 92 min (u = 105.4 and 107.0, l = 78.0 and 79.2) calves had greater ($p \leq 0.01$) lying bout duration than C-R8 $\mu = 64$ and 58 min (u = 74.5 and 68.4, l = 55.2 and 50.6) calves on d 3 and d 5, respectively. Previous studies have shown that N calves are less fit for transport compared to C calves due to stress associated with weaning, as well as feed, and feed bunk adaptation [5, 7] and therefore are more likely to lay down for longer periods of time due to exhaustion. In the current study, N calves had greater standing and lying bout durations compared to C calves. As previously mentioned, greater standing bout duration could be associated with the fact that N calves may be more restless due to weaning and greater lying bouts could be due to N calves not being used to the feed and the feed bunk. Interestingly, N calves had greater lying bout duration than C calves on d 2, however, differences observed for lying bout duration on d 3 and 5 were limited to animals that received an 8 h rest. This could be due in part, to the delay in R8 calves reaching their final destination as a result of the 8 hour rest.

Overall, standing and lying behaviour differences were observed across treatments. It is difficult to explain the treatment differences observed prior to calves being exposed to their rest and post-rest transport durations and therefore caution should be taken when interpreting these results. In addition, two calves from the C-R0-T15 group lost their accelerometers during transportation. Although we do not see clear differences between this particular treatment group and other groups, the reduced number of animals per treatment (10 vs 12) could have affected the results.

## Attitude and lameness score

Sampling point key: animals were assessed after the 20 h transport (UN1) and after the additional 4 or 15 h transport (UN2).

Attitude and lameness scores were assessed in order to account for potential injuries or a decrease in calf vigour due to multiple loading and unloading events. Lameness scoring has been previously used in transport studies assessing cull beef cows transported under Canadian winter conditions [23]. No differences ($p > 0.10$) were observed for attitude or lameness score after the initial 20 h transport (UN1) or after the additional 4 or 15 h transport (UN2). Attitude scores ranged between 0 (n = 323) and 1 (n = 5) at UN1 and between 0 (n = 313) and 1 (n = 13) at UN2. Lameness scores ranged between 0 (n = 326) and 1 (n = 2) at UN1 and 0 (n = 320), 1 (n = 5), and 2 (n = 1) at UN2. These results were similar to a previous study where the majority of calves (97%) had an attitude and lameness score of 0 [5]. Lack of differences observed in attitude and lameness score in the previous study were explained by the fact that all calves were conditioned calves, and therefore were more fit for transport [5]. Contrary to our results, a previous study observed greater attitude scores at UN2 compared to UN1 for R8 calves, and greater attitude scores in N than C calves sourced directly from the ranch [7]. Although differences were observed between N and C calves as well as UN2 compared to UN1 in R8 calves, differences were small (0.20–0.23) and may lack biological significance [7]. Based on the previous findings, the results observed in the present study were expected as calves were transported for a shorter period of time. Small or lack of differences observed across studies could be due to cattle being 'stoic' animals which do not overtly display fatigue or pain. Attitude and lameness scores may be a more valuable tool to use when assessing high risk cattle (e.g. market cows).

## DMI and feeding behaviour

Treatments key: conditioned (C) and non-conditioned (N) calves, rested for 0 (R0) or 8 (R8) h and transported for and additional 4 (T4) or 15 (T15) h of transport after rest.

A conditioning × time (nested in rest) effect ($p < 0.01$) was observed for DMI, where C-R0 calves had greater ($p < 0.01$) DMI than N-R0 calves and C-R8 calves had greater ($p \leq 0.05$) DMI than C-R0 and N-R8 calves on d1 (Fig 4A). Both C-R0 and C-R8 calves had greater ($p \leq 0.02$) DMI than N-R0 and N-R8 calves, respectively, on d 2 and 3. These results are similar to those obtained with the GrowSafe individual feed intake system. Data from GrowSafe showed that C calves had larger and more frequent meals, and the rate of feeding and intake were greater, and the time spent feeding was longer than N calves on d 1 to 4 after transport (S1 File). Although, data obtained from the GrowSafe system was only collected from one pen per treatment, similar results were reported in a previous study where C calves had greater DMI than N calves on d 0, 1, and 2 after transport [7]. The C calves were adapted to the feed and the feed bunk 28 d prior to transport and therefore we expected to have greater DMI than N calves. Differences of this kind were reported by several other studies in beef cattle [24–26].

During the 8 h rest period individual feeding behaviour was assessed for a subset of pens (1 pen/treatment with 10 animals/pen). However, data analysis was not possible due to incomplete

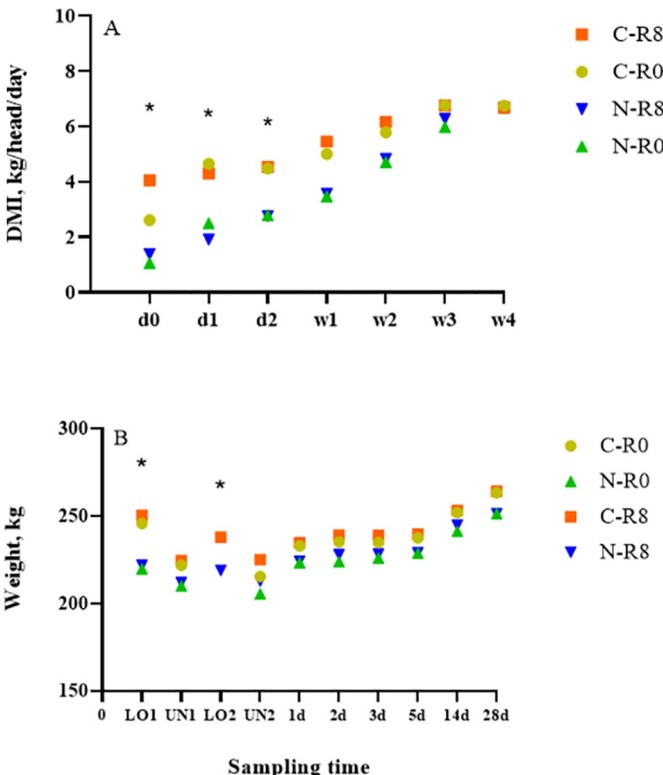

**Fig 4. Least squares-means (± upper and lower limits at 95% confidence) of (A) dry matter intake (DMI) and (B) body weight of conditioned (C) and non-conditioned (N), calves rested for 0 (R0) or 8 (R8) h and transported for an additional 4 (T4) or 15 (T15) h.** *Indicates significant differences between treatments ($p < 0.05$).

data sets. The GrowSafe system detected 19 out of 20 calves from the conditioned group, while only 3 out of 20 calves from the non-conditioned group. The low number of animals detected in the non-conditioned pens could be due to non-conditioned calves not visiting the feed bunk because they were unfamiliar with the pens, because calves were eating the bedding (straw), or due to a technical problem at the time of reading the ear tags. Unfortunately, we are not able to confirm if this was the case as we did not video record the animals during rest.

## Body weight

A conditioning × time (nested in rest) effect ($p < 0.01$) was observed for mean BW, C-R0 calves had greater ($p < 0.01$) BW than N-R0 calves, and C-R8 calves had greater ($p < 0.01$) BW than N-R8 calves at LO1 (Fig 4B). At LO2, C-R8 calves had greater ($p < 0.01$) BW than N-R8 calves. Conditioned calves had been receiving a total mixed ration (TMR) diet with 65% to 85% corn silage, 10 to 20% alfalfa hay, 13% barley grain, and 2% supplement for 28 d prior to the start of the trial, while N calves were grazing and suckling while out on pasture. However, differences in weight were only observed at LO1 and LO2, contrary to a previous study were C calves had greater BW than N calves over the entire experimental period [7]. We expected to see similar BW differences between C and N calves, as the conditioning period in this study was 8 d longer than in Meléndez et al. [7]. Although no BW differences were observed, C calves had numerically greater BW than N calves throughout the study. These findings are similar to previous studies that have reported greater BW in preconditioned calves than newly weaned calves [27–29].

## Shrink

A conditioning effect ($p < 0.01$) was observed for shrink after the 20 h transport (shrink1), where C calves shrank more ($p < 0.01$) than N calves. This is likely a result of C calves having greater gut fill than N calves. Conflicting shrink results have been reported in the literature for preconditioned and newly weaned calves. For example, greater shrink has been reported in conditioned compared to newly weaned calves after 15 and 36 h of transport [7, 25], while no shrink differences were observed between preconditioned and newly weaned calves after 26, 32, and 34 h or 64 to 241 km of transport [26, 30]. Differences between studies could be due to a series of factors such as weaning stress, transport duration, handling, space allowance, and weather [31].

A conditioning ($p < 0.01$) and a rest ($p < 0.01$) effect were observed for shrink after the additional 4 h transport, where C calves shrank 0.8% more ($p < 0.01$) than N calves, while R8 calves shrank 0.8% more ($p < 0.01$) than R0 calves. A conditioning × rest ($p < 0.01$) effect was also observed for shrink following the additional 15 h transport, where C-R8 calves shrank 3.0 and 2.7% more ($p < 0.01$) than C-R0 and N-R8 calves, respectively. The N-R8 calves shrank 0.9% more ($p < 0.01$) than N-R0 calves. These findings mimic what we observed for shrink following the initial 20 h transport because R8 calves had an opportunity to eat and drink during their rest period and C calves would have been more inclined to eat because they were familiar with the feed and the feed bunk. This has been confirmed by several other studies indicating that calves accustomed to the feed and the feed bunk had greater bunk visits than naive calves [32, 33]. Greater shrink observed in the present study for R8 and C calves could be due to calves having more gut fill to loose during transport. The results observed for R8 calves were contrary to previous studies where no shrink differences were observed between R8 and R0 calves after the additional 4 h transport [5, 7]. Although greater shrink was observed for R12 than R0, R4 and R8 calves, no differences were observed when adjusting for feed intake [5]. Differences between studies could be due to differences in initial transportation time (20 vs 36 h) or due to the different treatments used in each study.

## ADG

A transport × rest effect ($p = 0.02$) was observed for ADG during the first week after transportation where R8-T4 $\mu$ = 0.3 kg (u = 0.90, l = -0.25) calves had greater ($p < 0.01$) ADG than R8-T15 $\mu$ = -0.3 kg (u = 0.27, l = -0.88) calves. A conditioning effect ($p < 0.01$) was observed for ADG on week 1, where N $\mu$ = 1.5 kg (u = 2.03, l = 0.93) calves gained more weight ($p < 0.01$) than C $\mu$ = -1.5 kg (u = -0.93, l = -2.00) calves. No differences ($p > 0.10$) were observed for ADG during week 2. A rest and conditioning effect (both $p < 0.01$) were observed for ADG between d 14 and 28, where R0 $\mu$ = 0.9 kg (u = 1.14, l = 0.74) calves had greater ($p < 0.01$) ADG than R8 $\mu$ = 0.7 kg (u = 0.89, l = 0.59) calves, while C $\mu$ = 0.9 kg (u = 1.11, l = 0.73) calves had greater ($p < 0.01$) ADG than N $\mu$ = 0.7 kg (u = 0.91, l = 0.59) calves.

Transport time after rest had an effect on ADG during the first week after transport as R8-T4 calves had greater ADG than R8-T15 calves. Although not statistically significant, similar results were observed for R0-T4 $\mu$ = 0.02 kg (u = 0.60, l = -0.55) and R0-T15 $\mu$ = -0.02 kg (u = 0.54, l = -0.57). These results suggest that a longer transport period after a rest has a negative impact on ADG during the first week after transport.

The N calves had greater ADG than C calves on week 1 after transport, while the opposite was true between d 14 and d 28. This was expected as N calves started receiving a more energy-dense diet at the feedlot compared to the diet on pasture with the dams, while C calves received the same energy diet. However, contrary to ADG results, C calves had greater DMI during d 1 to 4 after transport compared to N calves. Greater ADG observed in N calves than

C calves during the first week after arrival to the feedlot could be due to the provision of a TMR ad-libitum diet. It has been reported that calves that are feed a low plane of nutrition and are subsequently fed a high plane of nutrition have a greater rate of gain than calves consistently fed a high plane of nutrition [34].

Conflicting results for ADG have been reported in the literature, where either no differences [25, 27], or greater ADG has been reported in newly weaned calves compared to preconditioned calves [26, 35]. Variation in results could be due to differences in pasture quality between and within experiments as well as how ADG was calculated. In the majority of studies, ADG was calculated for the experimental period, while in the present study ADG was calculated weekly for the first 2 weeks (week 1 and 2) and for the time between 14 and 28 d.

Calves in the present study were all weighed at 0800 on day 5, 14, and 28, therefore sampling time cannot explain the differences observed for ADG. As previously mentioned, difference observed in weight gain may be due to compensatory weight gain as newly weaned calves will increase weight when feed is changed from a low to a high energy diet [34]. However, a plateau was likely reached for N calves which can explain why C calves had greater ADG on d 14 to 28. The R0 calves had greater ADG than R8 between d 14 and 28 after transport. Differences observed between treatments could be due to transport stress resuming earlier in R0 calves compared to R8 calves, as R0 calves reached their 'final destination' 12 to 23 h sooner than R8 calves. However, no differences were observed for DMI between R0 and R8 calves. A previous study also reported greater ADG in C compared to N calves, and in R0 compared to R8 calves between d 14 and 28 [7], however, a different study reported no difference in ADG between R0, R4, R8, and R12 calves [5]. Inconsistencies between studies may be explained by the fact that ADG was assessed over a 28 d period in one of the studies [5] and weekly and biweekly [7] in the other. However, although no statistical differences were observed between calves rested for different periods of time, numerically R0 calves had greater or equal ADG than R4, R8, and R12 calves [5]. Overall, ADG showed consistent results across studies for rest and conditioning, where N calves had greater ADG in the first week after transport, while C and R0 calves had greater ADG between 14 and 28 d. Based on these results providing a rest was detrimental to ADG between 14 and 28 d after transport. This could be due to expediting the arrival to the final destination, which can also reduce the stress associated with handling and transport duration.

## Cortisol

Treatments key: conditioned (C) and non-conditioned (N) calves, rested for 0 (R0) or 8 (R8) h and transported for and additional 4 (T4) or 15 (T15) h of transport after rest.

Sampling point key: animals were samples before (LO1) and after the 20 h transport (UN1) and before (LO2) and after (UN2) the additional 4 or 15 h transport.

A conditioning × time (nested in rest) effect ($p < 0.01$) was observed for mean serum cortisol, where C-R0 and C-R8 calves had greater ($p \leq 0.04$) cortisol concentrations than N-R0 and N-R8 calves, respectively, at LO1, UN1, UN2, d 1, 2, and 3 (Fig 5A). In addition, the C-R8 calves had greater ($p = 0.01$) cortisol concentrations than N-R8 calves at LO2 and C-R0 calves had greater ($p = 0.03$) cortisol concentrations than N-R0 calves on d 5. Based on these results, C calves had greater cortisol concentrations than N calves. This result is contrary to previous studies, where no differences [7] or greater cortisol concentrations [25] were observed between newly weaned and conditioned calves prior to and after transport. Differences between the present study and the previous study [7] could be due to differences in the initial transport duration (20 vs 36 h) which may have allowed us to capture the cortisol peak due to a shorter transport duration. However, previous studies have reported plasma cortisol to peak at 4.5 and

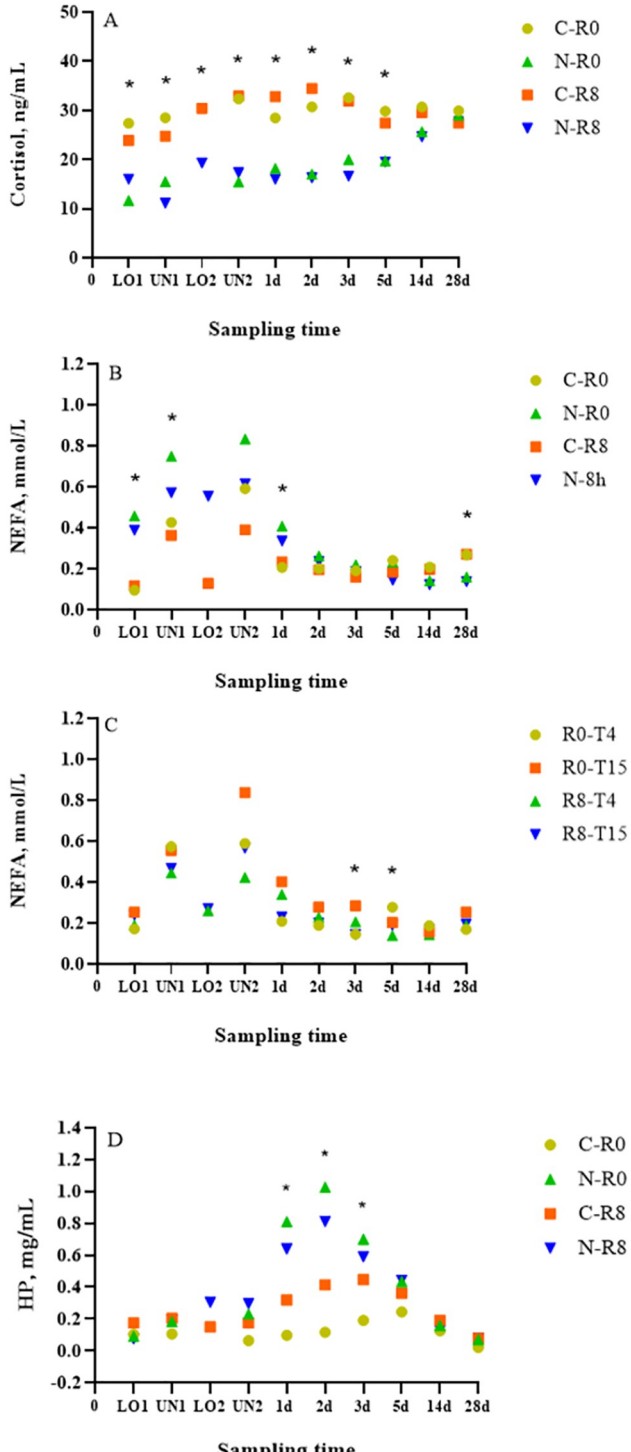

**Fig 5. Least squares-means of (A) cortisol, (B and C) non-esterified fatty acids (NEFA), and (D) haptoglobin (HP) of conditioned (C) and non-conditioned (N), calves rested for 0 (R0) or 8 (R8) h and transported for an additional 4 (T4) or 15 (T15) h.** *Indicates significant differences between treatments ($p < 0.05$).

12 h with a 9 and 31 h transport, respectively [20, 36] so it is likely that the peak plasma cortisol was missed in the present study. In addition, differences in cortisol concentrations were observed up to d 6, which was unexpected. Differences between studies could be due to differences in conditioning, where handling and transportation after 4 weeks of conditioning could potentially be more stressful than a period of 3 weeks, or that C calves were more excitable than N calves. Although all calves were sourced from two ranches with the same genetics, overall flight speed was 0.50 to 0.97 m/s greater in C calves than N calves. Differences observed in cortisol concentration could be due to higher temperament as flight speed has been reported to be correlated with greater plasma cortisol concentrations compared to calm cattle [37]. Differences observed in cortisol concentrations could also be due to N calves reaching a cortisol ceiling effect prior to the start of the trial due to weaning and/or transport stress, or due to differences in the time of day samples were collected as cortisol has a particular circadian rhythm.

Cortisol concentrations did not differ between R0 and R8 calves or between T4 and T15 calves. These results are contrary to previous studies assessing rest stops where greater cortisol concentrations were observed in calves that did not receive a rest compared to calves that received a rest [4] and in calves rested for 15 h compared to calves rested for 5 or 10 h [3]. Differences between Cooke et al. [4] and the current study may be explained by the different transport and rest times imposed in the studies. Calves that did not receive a rest were transported continuously for 1,290 km, while calves that received a rest were transported for a total of 1,290 km but provided a 2 h rest after every 430 km [4]. In our study, calves were transported for a total of 20 h, provided with an 8 h rest, and transported for an additional 4 or 15 h. Although no differences were observed for cortisol concentrations between calves provided a rest or not, our results are similar to our previous studies where no cortisol concentration differences were observed in calves rested for 0, 4, 8, and 12 h [5] or 0 and 8 h [7]. Overall, there was a lack of cortisol differences between rest treatments across this and the other two studies conducted by Meléndez et al. [5, 7] where the type of animals, distance, and rest periods were similar.

## NEFA

A conditioning × time (nested in rest) effect ($p < 0.01$) was observed for mean NEFA concentrations, where N-R0 calves had greater ($p \leq 0.03$) NEFA concentrations than C-R0 calves at LO1, UN1, and d1 (Fig 5B). The N-R8 calves had greater ($p < 0.01$) NEFA concentrations than C-R8 calves at LO2, while C-R8 calves had greater ($p \leq 0.03$) NEFA concentrations than N-R8 calves at d 28. NEFA concentrations rise as a result of fat mobilization due to feed deprivation [38]. Elevated NEFA concentrations seen in N calves at LO1, UN1, and d1 were expected, because non-conditioned calves had been without feed for a longer period of time prior to the 20 h transport as a result of being weaned and transported from their ranch of origin to the research facility where the study treatments were imposed. However, differences ($p \leq 0.03$) in NEFA concentrations were only observed between N-R0 and C-R0 calves but no differences ($p > 0.10$) were observed between N-R8 and C-R8 calves. We would have expected to see similar differences between N-R8 and C-R8 calves at LO1 and UN1 as these sampling points occurred before the provision of rest. Our findings are in agreement with Meléndez et al. [7] who reported greater NEFA concentrations in N-R8 compared to C-R8 calves at LO2. This is likely due to N calves being naive to the feed and the feed bunk, which could potentially affect feed intake and therefore NEFA concentrations [7, 25, 27]. Differences in NEFA concentrations observed at d 28 are hard to explain as NEFA concentrations have been reported to decrease rapidly after access to feed and water [39].

No differences ($p > 0.10$) were observed for NEFA concentrations between R0 and R8 calves. These results are contrary to results observed in previous studies, where 36-R0 calves

had greater NEFA concentrations than 36-R4 and 36-R8 calves at UN2 [5], while C-R0 calves had greater NEFA concentrations than C-R8 calves at UN2 [7]. In the present study, we would have expected to see greater NEFA concentrations in C-R0 than C-R8 calves because C calves had been adapted to the feed and the feed bunk and therefore C-R8 calves would likely eat during the 8 h rest period, which would decrease NEFA concentrations. In addition, and according to feed intake described above, the lack of differences between N-R0 and N-R8 calves observed in the present study could be due to N calves not being familiar with the feed and feed bunk [7, 24, 26], therefore calves would be less likely to eat during the 8 h rest. Differences in NEFA concentrations observed between studies could be due to differences in transportation time, as calves in the present study were transported for 20 h of transport, while in the previous studies calves were transported for 36 h [5, 7]. A shorter transport duration could result in lower NEFA concentrations, as calves are feed deprived for a shorter period of time. Differences in NEFA concentrations have been observed in previous studies with different transport durations, where differences were observed between R0, R4, and R8 calves transported for 36 h, however, no differences were observed between R0, R4, R8 and R12 calves transported for 12 h [5].

A transport × time (nested in rest) effect ($p < 0.01$) was observed for mean NEFA concentrations, where R0-T15 $\mu$ = 0.3 mmol/L (u = 0.38, l = 0.21) calves had greater ($p = 0.01$) mean NEFA concentrations than R0-T4 $\mu$ = 0.1 mmol/L (u = 0.19, l = 0.11) calves on d 3, while R0-T4 $\mu$ = 0.3 mmol/L (u = 0.37, l = 0.21) calves had greater ($p < 0.01$) mean NEFA concentrations than R0-T15 $\mu$ = 0.2 mmol/L (u = 0.27, l = 0.16) calves on d 5 (Fig 5C). Calves transported for a longer period of time would be expected to have greater NEFA concentrations at UN2 instead of on d 3. In addition, the results are contradictory as on d 5, NEFA concentrations were greater in T4 compared to T15 calves. Although statistically significant, results may lack biological relevance as NEFA concentrations prior to feed deprivation have been reported between 0.2 and 0.3 mmol/L in beef cattle [39], similar to NEFA concentrations observed on d 3 and 5.

## HP and SAA

HP and SAA increase in cases of infection, inflammation, or physical trauma and peak 2 to 3 days after the stimuli [40]. A conditioning × time (nested in rest) effect ($p < 0.01$) was observed for HP, where N-R0 calves had greater ($p < 0.01$) mean HP concentrations than C-R0 calves on d 1 and 3 (Fig 5D). Two days post-transport, N-R0 and N-R8 calves had greater ($p < 0.01$) mean HP concentrations than C-R0 and C-R8 calves, respectively. Similar to our results, greater HP concentrations have been reported in N than C calves at UN1, d 1, 2, 3, and 6 [7].

A conditioning × time (nested in rest) effect ($p < 0.01$) was also observed for mean SAA, where N-R0 calves had greater ($p < 0.01$) SAA concentrations than C-R0 calves at UN2, d 1, 2, and 3, while N-R8 calves had greater ($p \leq 0.01$) SAA concentrations than C-R8 calves at LO2, UN2, d 1, and 2 (Fig 6A). A transport × time (nested in rest) effect ($p < 0.01$) was observed for SAA, where R0-T15 $\mu$ = 403 µg/mL (u = 596.6, l = 273.1) and R8-T4 $\mu$ = 318 µg/mL (u = 471.9, l = 215.3) calves had greater ($p \leq 0.02$) SAA concentrations than R0-T4 $\mu$ = 150 µg/mL (u = 223.5, l = 100.6) calves at UN2 (Fig 6B). The results observed for HP and SAA in the present study were above baseline concentrations previously reported for cattle (HP < 0.1 g/L and SAA 1.3 µg/mL) [41]. Similar results were reported in a previous study, where greater HP and SAA concentrations were observed in N compared to C calves on d 1, 2, and 3 [7]. Greater HP and SAA concentrations in N calves could be a result of stress caused by weaning, as both markers have been reported to increase 3 to 5 d after weaning [42]. Increases in acute phase

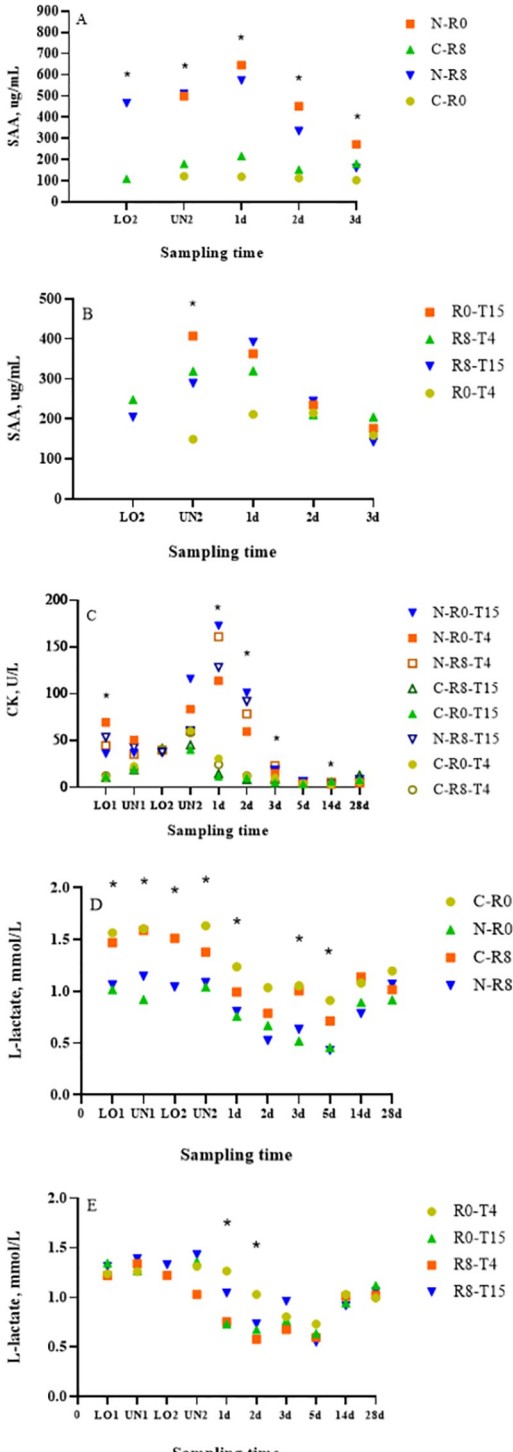

**Fig 6. Least squares-means of (A and B) serum amyloid-A (SAA), (C) creatine kinase (CK), and (D and E) L-lactate concentrations of conditioned (C) and non-conditioned (N), calves rested for 0 (R0) or 8 (R8) h and transported for an additional 4 (T4) or 15 (T15) h.** *Indicates significant differences between treatments ($p < 0.05$).

proteins such as SAA and HP have been suggested to be a result of direct or indirect activation of IL-1 and IL-6 associated with increased circulating corticosteroids [43]. However, this is in contrast to our current findings where greater cortisol concentrations were observed in C than N calves indicating that acute phase proteins are being activated by a non-corticosteroid dependant pathway.

Contrary to the previous study [7] where no differences were observed for HP or SAA concentrations between R0 and R8 calves, R8-T4 calves in the present study had greater SAA concentrations than R0-T4 calves at UN2. This result is difficult to explain because R8-T4 calves received an 8 h rest prior to the additional transport that the R0-T4 group did not receive. Furthermore, R0-T15 calves had greater SAA concentrations than R0-T4 calves at UN2. This was expected as R0-T15 calves were transported for 11 more h than R0-T4 calves. Interestingly, the previously mentioned differences observed for rest and transport duration after rest only occurred at UN2 which is also difficult to explain as SAA and HP concentrations remain elevated for 2–3 d after the stimuli, as was seen in N calves.

## CK

A conditioning × transport × time (nested in rest) effect ($p$ = 0.02) was observed for mean CK. At LO1, d 1 and 2, the N-R0-T4, N-R8-T4, and N-R8-T15 calves had greater ($p \leq 0.04$) CK concentrations than C-R0-T4, C-R8-T4, and C-R8-T15 calves, respectively (Fig 6C). On d1 and 2, the N-R0-T15 calves had greater ($p < 0.01$) CK concentrations than C-R0-T15 calves while on d 3, N-R0-T15 and N-R8-T4 calves had ($p \leq 0.02$) CK concentrations than C-R0-T15 and C-R8-T4 calves, respectively. CK is an enzyme involved in the production of ATP in the muscle which appears in the plasma after muscle damage [44]. Therefore, CK has been used as an indicator of muscle fatigue in transportation studies whose concentrations have been shown to increase progressively with increasing transportation duration [45]. Similar findings were reported in a previous study, where N calves had greater CK concentrations than C calves at LO1, d 1, 2, and 3 [7]. The authors speculated that the greater CK concentrations observed in N calves could be explained by greater physical activity associated with the combined effects of increased physical activity [46] such as gathering of the cow-calf pairs at the ranch, weaning, loading, unloading, and transport. These results are consistent with those observed for HP, SAA, and NEFA concentrations, where N calves have greater concentrations than C calves, suggesting that conditioned calves are more fit for transport than their non conditioned counterparts [5, 7].

## L-lactate

A conditioning × time (nested in rest) effect ($p < 0.01$) was observed for mean L-lactate concentrations. The C-R0 calves had greater ($p \leq 0.05$) L-lactate concentrations than N-R0 calves at LO1, UN1, UN2, d1, 2, 3, and 5 (Fig 6D). The C-R8 calves had greater ($p \leq 0.05$) L-lactate concentrations than N-R8 calves at LO1, UN1, LO2, and d 3. Lactate has been measured in transport studies as an indicator of muscle damage [44]. These findings are contrary to expected, as N calves were handled more as a result of being gathered for weaning and transportation from the ranch of origin to the research centre. Therefore, L-lactate concentrations in N calves should have been elevated because they experienced greater combined physical activity. However, it may be possible that gathering the cattle and transport prior to 20 h transport did not result in enough physical exertion to see a difference in lactate concentrations between treatments. Similar results were reported in a previous study where C calves had greater lactate concentrations than N calves at UN1 and d 14 [7]. Similar results were observed for cortisol concentrations where C calves had greater concentrations, however they are contrary to the results observed for SAA, HP, NEFA, and CK.

A transport × time (nested in rest) effect ($p$ = 0.02) was observed for mean L-lactate, where R0-T4 calves had greater ($p$ = 0.02) L-lactate concentrations than R0-T15 and R8-T4 calves on d 1 after transport (Fig 6E). The R0-T4 calves had greater ($p$ = 0.02) L-lactate concentrations than R8-T4 calves on d 2 after transport. Greater L-lactate concentrations observed for R0 calves compared to R8 calves were expected since animals receiving a rest should be less fatigued than animals that did not receive a rest. However, we would have also expected to see similar differences between R0-T15 and R8-T15 calves. Greater L-lactate concentrations observed in T4 compared to T15 calves cannot be explained since T4 calves were transported for a shorter period of time.

## Complete blood cell count

A conditioning × time (nested in rest) effect ($p$ < 0.01) was observed for mean HCT. The N-R8 $\mu$ = 34% (u = 35.3, l = 32.1) calves had greater ($p$ = 0.03) HCT than C-R8 $\mu$ = 31% (u = 32.2, l = 29.4) calves at LO1. On d 28, the C-R0 $\mu$ = 29% (u = 30.7, l = 28.1) and C-R8 $\mu$ = 30% (u = 31.1, l = 28.4) calves had greater ($p \leq$ 0.02) HCT than N-R0 $\mu$ = 27% (u = 28.1, l = 25.6) and N-R8 $\mu$ = 27% (u = 27.8, l = 25.4) calves, respectively. These results differ from previous studies where no differences were observed in HCT between calves that were C and N as well as rested and not rested [5, 7]. The results observed at LO1 could be explained by the fact that N calves were unable to eat or drink for a longer period of time prior to LO1 than C calves. Interestingly, similar differences between C-R0 and N-R0 calves were not observed. We expected to see differences in HCT after long periods of feed and water deprivation like after the 20 h (UN1) or the 15 h (UN2) transport. However, no differences have been previously reported in HCT after 12 or 36 h of transport [5, 7]. It is challenging to estimate the level of dehydration in cattle based on the time animals last had access to feed or water, because ruminants can redirect water contents from the rumen to the main circulation [44]. The results observed for d 28 are puzzling, as HCT should return to normal 48 hours after rehydration [47]. Although statistically significant, differences observed for HCT were 2 to 3%, which may not be biologically relevant.

A transport × time (nested in rest) effect ($p$ < 0.01) was observed for WBC, where R8-T15 $\mu$ = 10×10³/μl (u = 11.8, l = 9.5) calves had greater ($p$ < 0.01) WBC counts than R8-T4 $\mu$ = 7×10³/μl (u = 8.8, l = 7.0) calves on d 5, while R8-T15 $\mu$ = 10 ×10³/μl (u = 11.4, l = 9.1) calves had greater ($p$ < 0.01) mean WBC counts than R8-T4 $\mu$ = 8.1×10³/μl (u = 9.1, l = 7.2) calves on d 28. Although statistically different, WBC counts were within the normal range for (4–12 × 10³/μl) beef calves [48]. White blood cell counts have been reported to remain unchanged after weaning but to increase after transportation [49].

A conditioning × time (nested in rest) effect ($p$ < 0.01) was observed for granulocyte counts. At UN1, N-R8 calves had greater ($p$ = 0.02) granulocyte counts than C-R8 calves. On d 1, N-R0 calves had greater ($p$ = 0.03) granulocyte counts than C-R0 calves. This was expected as N calves were exposed to more stressors than C calves prior to transport. Similar results were reported in a previous study where N calves had greater granulocyte counts than C calves on d 1 and 2 post transport. A transport × time (nested in rest) effect ($p$ = 0.02) was observed for granulocyte counts. On d 5, R8-T15 calves had greater ($p$ < 0.01) granulocyte counts than R8-T4 calves. This could be because calves that were transported longer could be more stressed than calves that were transported for 4 hours, however we would expect to see similar differences for R0-T15 and R0-T4 calves. Although statistical differences were observed, granulocyte counts were within the normal range (1.9–7.9 × 10³/μl) for beef cattle [48].

## Morbidity and mortality

Over the 28-d experimental period, morbidity and mortality rates were 7% and 0%, respectively. A total of 25 animals were treated due to fever (C-R0-T4 = 3; C-R0-T15 = 1; C-R8-T4 = 9;

C-R8-T15 = 6; N-R0-T4 = 4; N-R0-T15 = 0; N-R8-T4 = 0; N-R8-T15 = 2). Morbidity % varied among our three studies (study 1: 2.5% [5], study 2: 5.9% [7], and present study: 7%). The lowest morbidity percentage was observed in our first study [5], which is likely due to the fact that all calves were conditioned. Surprisingly, the majority of animals treated in the second (C = 13 and N = 6) [7] and present (C = 19 and N = 6) study were C calves. Numerical differences could be due to timing of prophylactic antibiotic administration as C calves received an antibiotic 28 d prior to the start of the trial, while N calves received an antibiotic during the trial (UN2). However, studies have shown that preconditioned calves have reduced morbidity (25 to 64% less) and mortality (10% less) compared to newly weaned calves [29, 50, 51]. Increased morbidity observed across studies could be a result of several factors, however, if we look at the additional treatments in the second [7] and present study we observe that an increased number of directly sourced calves (11) were treated compared to auction market (7) calves, while an increased number of calves that were transported for 4 h (16) were treated compared to calves that were transported for 15 h (9). Morbidity in the second [7] and present study were approximately double and triple the morbidity reported in the first study [5]. As previously mentioned, the low morbidity observed in the first study could be due to all calves being preconditioned. Differences in morbidity observed between the present and previous study [7] could be due to differences in treatments (4 vs 15 h post-rest transport duration) and conditioning period (20 vs 28 d). Interestingly, although not statistically significant, there was a greater number of calves treated in treatment groups that had shorter transport durations and less handling. Morbidity was not statistically different between R0 and R8 calves but it was numerically greater in R8 (18) than R0 (8) calves in the present study. These results are similar to our previous studies where numerically greater morbidity was observed in R12 (6) calves than R8 (0), R4 (1), and R0 (1) calves [5], and in R8 (11) compared to R0 (8) calves. These findings should be viewed with caution due to the very low rates of morbidity and mortality in this study, which could not be analyzed statistically. The lack of mortality observed across studies (0%) could be due to the limited number of animals assessed and due to a short trial duration (28 d) even though it is the period of time that receiving calves would normally be diagnosed with BRD [52].

## Conclusions

Overall, few and inconsistent differences were observed between rest treatments, where rest improved DMI and L-lactate concentrations, and negatively affected standing % and ADG. Few differences were observed for transport duration after rest, where shorter transport had greater ADG and lower WBC and granulocytes counts. Rest did not seem to be more beneficial when the post-rest transport duration was longer. In addition, the N calves had greater physiological and behavioural indicators of reduced welfare than C calves. Based on these results, the best way to improve cattle welfare is to condition animals as early as 21 d prior to transport. Future studies should assess which aspects of conditioning (i.e. weaning, painful procedures, vaccination) are most impactful such that labour and costs associated with conditioning are reduced but remain effective at improving cattle welfare.

## Supporting information

**S1 File. Detailed feeding behavior results and least squares-means of feeding behavior for conditioned (C) or non-conditioned (N) calves rested for 0 (R0) or 8 (R8) h and transported for an additional 4 (T4) and (T15) h.** (A) Meal duration, (B) meal frequency, (C) meal size, (D) feeding intake, (E) feeding rate, and (F) feeding time.
(DOCX)

**S1 Table. Generalized linear mixed modelling (SAS POC GLIMMIX statements) indicating the response variable, the selected distribution, the link function, and the selected structure of the covariance matrix.**
(DOCX)

**S2 Table. Least squares-means (± upper and lower limits at 95% confidence) of production and behavioural parameters of conditioned (C) and non-conditioned (N) calves rested for 0 (R0) or 8 (R8) h and transported for an additional 4 (T4) and (T15) h.**
(DOCX)

**S3 Table. Least squares-means (± upper and lower limits at 95% confidence) of physiologic parameters of conditioned (C) and non-conditioned (N) calves rested for 0 (R0) or 8 (R8) h and transported for an additional 4 (T4) and (T15) h.**
(DOCX)

**S1 Raw data.**
(XLSX)

## Acknowledgments

The authors appreciate the invaluable help of Agriculture and Agri-Food Canada research feedlot staff and beef welfare technicians Nicholaus Johnson and Dawn Gray and master student Nicholas Wong. We would also like to thank Nelson Family Ranches for their participation in this project and Alberta Prime for transporting and caring for the calves used in this study.

## Author Contributions

**Conceptualization:** Daniela M. Meléndez, Sonia Marti, Karen S. Schwartzkopf-Genswein.

**Data curation:** Daniela M. Meléndez.

**Formal analysis:** Timothy D. Schwinghamer, Xiaohui Yang.

**Funding acquisition:** Derek B. Haley, Karen S. Schwartzkopf-Genswein.

**Investigation:** Daniela M. Meléndez, Karen S. Schwartzkopf-Genswein.

**Methodology:** Daniela M. Meléndez, Sonia Marti, Karen S. Schwartzkopf-Genswein.

**Project administration:** Daniela M. Meléndez, Sonia Marti, Karen S. Schwartzkopf-Genswein.

**Resources:** Derek B. Haley, Karen S. Schwartzkopf-Genswein.

**Software:** Daniela M. Meléndez.

**Supervision:** Sonia Marti, Karen S. Schwartzkopf-Genswein.

**Validation:** Daniela M. Meléndez.

**Visualization:** Daniela M. Meléndez.

**Writing – original draft:** Daniela M. Meléndez.

**Writing – review & editing:** Daniela M. Meléndez, Sonia Marti, Derek B. Haley, Timothy D. Schwinghamer, Xiaohui Yang, Karen S. Schwartzkopf-Genswein.

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
