## [Decision Letter · Decision Letter 0]

20 Jul 2022

PONE-D-22-13599Effect of rest, post-rest transport duration, and conditioning on performance, behavioural, and physiological welfare indicators of beef calvesPLOS ONE

Dear Dr. Meléndez,

Thank you for submitting your manuscript to PLOS ONE. After careful consideration, we feel that it has merit but does not fully meet PLOS ONE’s publication criteria as it currently stands. Therefore, we invite you to submit a revised version of the manuscript that addresses the points raised during the review process.

We look forward to receiving your revised manuscript.

Kind regards,

Marcio Duarte, PhD

Academic Editor

PLOS ONE

Journal Requirements:

    "This project was funded by the Beef Cattle Research Council (http://www.beefresearch.ca/) and Agriculture and Agri-Food Canada (http://www.agr.gc.ca/eng/home/?id=1395690825741) Sustainable Beef and Forage Cluster under the Canadian Agricultural Partnership AgriScience Program (ANH.06.17 AIP-CL01). The co-author Sonia Marti was partly supported by the CERCA program from Generalitat de Catalunya. "

Additional Editor Comments:

At first, please accept my apologies for the delay in getting the manuscript first decision. As you may know, find good reviewers with time to contribute has been a major issue for Editors at all journals. Just for you notice, a total of 10 reviewers were invited to review the manuscript before two accepted the task. As an Editor, I think it is important to bring that information up in respect to the authors to help not discourage submitting to PLoS one in the future.

I look forward to receive the revised version of the manuscript at your earliest convenience. 

Reviewers' comments:

Reviewer's Responses to Questions

**Comments to the Author**

1. Is the manuscript technically sound, and do the data support the conclusions?

Reviewer #1: Yes

Reviewer #2: Partly

2. Has the statistical analysis been performed appropriately and rigorously? 

Reviewer #1: Yes

Reviewer #2: No

3. Have the authors made all data underlying the findings in their manuscript fully available?

Reviewer #1: Yes

Reviewer #2: No

4. Is the manuscript presented in an intelligible fashion and written in standard English?

Reviewer #1: Yes

Reviewer #2: No

5. Review Comments to the Author

Reviewer #1: Line 68 – spell out verses

Line 350 – 351 – Could there be a weather by travel time interaction for standing?

Line 362 – 364 – What were the pen conditions? Wet? Different temperature? Cattle standing or lying could be determined by these. With differences in hauling time, could pen conditions be different??

Line 372 – 373 – This sentence needs rewritten. ……mean standing percentage, R0 calves had greater…. Maybe make this two sentences or drop the beginning. Or place the word “as” between percentage and R0 calves.

Lines 382 – 386 – Difference in time of day but also in difference in THI or weather??

General question – What does standing mean? It seems very important in this study. However, we find docility, floor temp, floor moisture, bedding and others have significant impacts on cattle standing or lying. Are we trying to read too much into this?

Could you re-analyze your docility/exit pace and correlate to standing?

I think the DMI and feeding behavior is much more diagnostic to fatigue and behavior than standing. This is excellent data. One question I have on intakes is about variation. Is there greater variation in N verses C treatment cattle? Were there N treatment cattle that did not eat or were more delayed on eating feed than cattle on C? We have seen this in dairy calves and it is very diagnostic on subsequent performance.

Lines 587 – 593 – Would you classify N cattle as feed restricted?? Not me. Was it due to gut fill? Also, this seems counter to your DMI data where C cattle consumed more feed in the first 4 days. Did you weigh all the cattle at the same time of the day?

615 – 617 – How can you contribute ADG to expedited stress but you did not see the same results for intakes?

With increasing CK and HP, do you think there was an underlying illness boiling in these calves? I would have liked to see result at 60 to 90 days out. Our practice sees a lot of cattle. Our data shows that ranch fresh cattle break with BRD at 28 to 42 days on feed. Some of the most interesting data might start where this study ended. This might explain your morbidity differences.

Also, C cattle would be more used to people interaction and potentially show clinical signs due to trust more than N cattle.

I did not see the flight speed data in table 2 as indicated in the foot note.

Reviewer #2: General comments:

This manuscript addresses a very important topic, seeking to analyse the effects of long-distance transport in conditioned or unconditioned calves with or without a resting period during transport on animal welfare. However, the study, developed with a complex experimental design, brought results difficult to interpret. It is aggravating the fact that the description of the experimental design is confusing, which makes understanding the text even more difficult, for example, in line 301 it is informed that time was nested at rest in the statistical model and further on (L336) it is informed that rest was nested in time. Furthermore, in certain parts of the text there is a disconnection between the sub-item and what follows, for example, the content of paragraph L137-139 is not related to “unconditioned calves”. The fact that many of the results of the present study are contradictory to those of previous studies carried out by the same research group or difficult to explain (L365, L481, L736 and L741) deserves special consideration. Additionally, most of the results are inconclusive. I had difficulty time when reading this manuscript, which in addition to being very long and with many variables (which resulted in many abbreviations), has several paragraphs that are not well written, making understanding very difficult. Thus, in my opinion this manuscript is not ready for publication.

Additional comments

Introduction

L45-67. Too long paragraph.

L66. Conditioning. This word has a specific meaning in behavioural studies (conditioning = “a behavioral process whereby a response becomes more frequent or more predictable in a given environment as a result of reinforcement, with reinforcement typically being a stimulus or reward for a desired response”, Encyclopaedia Britannica, 2022). Its use in the context of this manuscript, despite being usual, is confusing. Think about and consider replacing it.

Materials and methods

L73. Replace “protocol” for “study”.

L76-77. Please, consider summarizing all this information here to facilitate reading.

L141-151. Very confusing paragraph.

L154-156. It would be good to have a Figure showing the distribution of these compartments in the trailer.

L164. Replace “Trailer temperature and humidity…” for “Air temperature and humidity within the trailers...”.

L175. Replace “Trailer temperature…” for “Air temperature…”.

L199. Again, you are using a complex word, “attitude”, when carrying out behavioural studies. Think about and consider replacing for alertness or other word less complicate.

L246. Sound strange to include “weight” as a physiological trait

336-351. Very hard to follow.

Results and discussion.

L356-358. Confusing, clarify.

L358-360. Remove

L365-367. Triple interactions are usually hard to explain.

…

6. PLOS authors have the option to publish the peer review history of their article (what does this mean?). If published, this will include your full peer review and any attached files.

Reviewer #1: No

Reviewer #2: No

---

## [Author Response · Author response to Decision Letter 0]

9 Sep 2022

Lethbridge, September 8th, 2022

Dear Dr. Marcio Duarte,

Thank you very much for your message of August 10th, 2022, regarding our manuscript PONE-D-22-13599 “Effect of rest, post-rest transport duration, and conditioning on performance, behavioural, and physiological welfare indicators of beef calves”. We are grateful for the constructive comments of the reviewers. We hereby would like to submit the revised version of our manuscript for publication in PLoS One. 

Please find our detailed response below, with our responses to the reviewers’ comments in blue text. Changes in the manuscript have been highlighted in red and blue.

Yours sincerely,

Daniela Meléndez

Comments to the Author

1. Is the manuscript technically sound, and do the data support the conclusions?

Reviewer #1: Yes

Reviewer #2: Partly

2. Has the statistical analysis been performed appropriately and rigorously? 

Reviewer #1: Yes

Reviewer #2: No-Please let us know in which way the statistical analysis has not been performed appropriately. 

3. Have the authors made all data underlying the findings in their manuscript fully available?

Reviewer #1: Yes

Reviewer #2: No Please let us know why you don’t consider the findings of the manuscript are fully available so that we may correct this. 

4. Is the manuscript presented in an intelligible fashion and written in standard English?

Reviewer #1: Yes

Reviewer #2: No

5. Review Comments to the Author

Reviewer #1: Line 68 – spell out verses

Line 68. Versus was spelled out as suggested. 

Line 350 – 351 – Could there be a weather by travel time interaction for standing?

The authors did not include weather in the model due to the number of covariates in the model (breed, group, flight speed) and due to the fact that the THI in the truck was not substantially different between weeks of transport. 

Line 362 – 364 – What were the pen conditions? Wet? Different temperature? Cattle standing or lying could be determined by these. With differences in hauling time, could pen conditions be different??

This is a good point. Thank you for your comment. All pens were muddy so all pens were provided with straw. Calves were rested at the same time so the pen conditions would have been the same. We have added a sentence in the text to clarify this (Line 378-379). 

Line 372 – 373 – This sentence needs rewritten. ……mean standing percentage, R0 calves had greater…. Maybe make this two sentences or drop the beginning. Or place the word “as” between percentage and R0 calves.

Lines 382-383. Thank you for your comment. We changed it to two sentences as we believe the beginning of the sentence is important to let the reader know that there was a significant time (nested in rest) effect. 

Lines 382 – 386 – Difference in time of day but also in difference in THI or weather??

Line 395. Thank you for your comment. We have added THI as a possible explanation for the difference in standing. 

General question – What does standing mean? It seems very important in this study. However, we find docility, floor temp, floor moisture, bedding and others have significant impacts on cattle standing or lying. Are we trying to read too much into this?

Standing is only one of the behavioural parameters that we collected. This section is more extensive than other sections in this study or in previous studies due to the number of interactions observed and due to the way in which standing behaviour was analyzed hourly during transport and rest. Although there are interactions that we cannot explain (e.g. Line 365-369), we believe it is important to report all the statistical differences observed between groups of interest. In the text we described the results but we don’t believe we are placing a lot of weight on standing alone based on our concluding remarks (Line 869-878). 

Could you re-analyze your docility/exit pace and correlate to standing?

We performed the analysis suggested. A Pearson’s correlation between flight speed and standing % of the pooled data showed a weak and not statistically significant (r = -0.03, p = 0.14) correlation. When divided into the different hours of transport correlations were as follows:

Hour of transport r p-Value

1 0.08 0.40

2 0.12 0.25

3 0.07 0.45

4 -0.04 0.67

5 -0.02 0.77

6 0.05 0.61

7 0.00 0.97

8 -0.04 0.64

9 -0.06 0.53

10 -0.05 0.56

11 -0.17 0.09

12 -0.07 0.44

13 -0.02 0.81

14 -0.01 0.87

15 -0.07 0.49

16 -0.08 0.43

17 -0.06 0.51

18 -0.09 0.37

19 -0.06 0.51

20 -0.10 0.29

I think the DMI and feeding behavior is much more diagnostic to fatigue and behavior than standing. This is excellent data. One question I have on intakes is about variation. Is there greater variation in N verses C treatment cattle? Were there N treatment cattle that did not eat or were more delayed on eating feed than cattle on C? We have seen this in dairy calves and it is very diagnostic on subsequent performance.

Thank you for your comment. We did see lower number of N calves eating during the rest period. This is mentioned in Lines 535-542. We believe that this may have been because calves were unfamiliar with the feed bunks and/or because N calves were eating the straw that was provided as bedding. We will add the straw part to the manuscript as we did not include it previously (Line 540). 

Variation observed between C and N calves

Average ± SD for DMI feed refusals:

 C calves 5.3 ± 1.42 kg/h/d 

 N calves 3.9 ± 1.99 kg/h/d

Average ± SD from GrowSafe feed intake during the first week after transport:

C calves 6.1 ± 1.98 kg/d 

 N calves 3.4 ± 1.72 kg/d

Average ± SD from GrowSafe feed intake during the first 28 days after trasport:

C calves 7.3 ± 2.01 kg/d

N calves 6.5 ± 2.50 kg/h/d

Lines 587 – 593 – Would you classify N cattle as feed restricted?? Not me. Was it due to gut fill? Also, this seems counter to your DMI data where C cattle consumed more feed in the first 4 days. Did you weigh all the cattle at the same time of the day?

Thank you for your comment. We agree that N calves were not be feed restricted but they had access to a lower energy diet while grazing which we were comparing to feed restriction. We understand that this is not an adequate comparison so we have changed the reference to a paper mentioning compensatory weight gain: ‘Cattle graziers have long noted that store cattle, wintered upon low planes of nutrition, gain in weight more rapidly when turned out to spring grass than do cattle which have been wintered on a higher plane of nutrition, by balanced feeding on concentrates’. 

Greater ADG observed in N calves during the first week after transport is a puzzling finding when you take into considerations that C calves had greater DMI during the first week of transport. It would be hard to explain this difference based on shrink as shrink due to gut fill is short lived and, as previously mentioned, the C calves had greater DMI than N calves during the first week after transport so gut fill would have been greater in C calves. 

Although this finding is unexpected, we observed similar results in our previous study. We speculate N calves have greater ADG during the first week after transport because their diet changed from a low-energy to a high-energy diet, while C calves received the same high-energy diet before and after transportation. As you can see in the graph below (just for illustration purposes), the ADG of cattle on feed is very steep during the first 20 days and then it starts to decrease. It could be that the N calves are at the beginning of this graph line while C calves are on the downward line as animals were conditioned for 28 days. Of course this is only speculation as we observe the opposite on the ADG between day 14 to d 28. 

(https://www.beefresearch.ca/topics/optimizing-feedlot-efficiency/)

Yes, cattle were all weighed at the same time on day 5, 14, and 28. This information has been added in Lines 614-615 to clarify.

615 – 617 – How can you contribute ADG to expedited stress but you did not see the same results for intakes?

We make a similar speculation with ADG between N and C calves. It could be possible that even though no differences are observed in DMI between R0 and R8 calves, stress caused by delayed transport could affect ADG. We will clarify that although differences were observed in ADG, no differences were observed for DMI between C and N (601-603) and between R0 and R8 calves (622-623). 

With increasing CK and HP, do you think there was an underlying illness boiling in these calves? I would have liked to see result at 60 to 90 days out. Our practice sees a lot of cattle. Our data shows that ranch fresh cattle break with BRD at 28 to 42 days on feed. Some of the most interesting data might start where this study ended. This might explain your morbidity differences.

Also, C cattle would be more used to people interaction and potentially show clinical signs due to trust more than N cattle.

Thank you for your comment. We agree that it would have been great to follow these cattle after 28 days. Cattle did remain in the research station and were assigned to different experiments. We do know that these cattle were overall healthy after the 28 day period and there was no mortality. However, we are not able to report morbidity after the experimental period because it can be affected by the experimental conditions of the trial they were assigned to. 

I did not see the flight speed data in table 2 as indicated in the foot note.

Thank you for your comment. We do not see where flight speed is mentioned in the footnote. We did not add flight speed to Table 2 because it was used as a covariate in the model and not described as a variable.

Reviewer #2: General comments:

This manuscript addresses a very important topic, seeking to analyse the effects of long-distance transport in conditioned or unconditioned calves with or without a resting period during transport on animal welfare. However, the study, developed with a complex experimental design, brought results difficult to interpret. It is aggravating the fact that the description of the experimental design is confusing, which makes understanding the text even more difficult, for example, in line 301 it is informed that time was nested at rest in the statistical model and further on (L336) it is informed that rest was nested in time.

Thank you for your comment. We apologize for the mistake. It should be time (nested in rest) as mentioned in the statistical section. We have changed this throughout the manuscript.

 Furthermore, in certain parts of the text there is a disconnection between the sub-item and what follows, for example, the content of paragraph L137-139 is not related to “unconditioned calves”. 

Thank you for your comment. We placed this section under sample collection Lines 189-191. 

The fact that many of the results of the present study are contradictory to those of previous studies carried out by the same research group or difficult to explain (L365, L481, L736 and L741) deserves special consideration. Additionally, most of the results are inconclusive. I had difficulty time when reading this manuscript, which in addition to being very long and with many variables (which resulted in many abbreviations), has several paragraphs that are not well written, making understanding very difficult. Thus, in my opinion this manuscript is not ready for publication.

We agree with the reviewer that some of the results in the present study are difficult to explain, however we don’t believe this is uncommon, especially in animal science. The manuscript is lengthy because we report all the significant interactions of interest that were observed. We believe this information is of interest to the reader and is worth reporting and discussing in order to best inform future studies. 

Additional comments

Introduction

L45-67. Too long paragraph.

Line 56. Thank you for your comment. We split the paragraph. 

L66. Conditioning. This word has a specific meaning in behavioural studies (conditioning = “a behavioral process whereby a response becomes more frequent or more predictable in a given environment as a result of reinforcement, with reinforcement typically being a stimulus or reward for a desired response”, Encyclopaedia Britannica, 2022). Its use in the context of this manuscript, despite being usual, is confusing. Think about and consider replacing it.

We understand that conditioning can mean something different. However, due to the use of this word in the previous two manuscripts, the authors would prefer to keep this word. We have also described conditioning in the manuscript (Line 126-127) so to reduce confusion with the definition you have provided. It should also be noted that preconditioning and conditioning are terms typically and widely used by cow calf producers, veterinarians and animal scientists to describe pre-shipping management of beef calves.

Materials and methods

L73. Replace “protocol” for “study”.

Line 73. Thank you for your comment. Protocol was replaced with study. 

L76-77. Please, consider summarizing all this information here to facilitate reading.

Line 77-79. The information was added as suggested. 

L141-151. Very confusing paragraph.

Line 142-154. The paragraph was split into two for the information of conditioned and non-conditioned calves to be on separate paragraphs to avoid further confusion. An additional words were added to clarify. 

L154-156. It would be good to have a Figure showing the distribution of these compartments in the trailer.

As suggested we have added a figure to depict the trailer compartments (Figure 2). 

L164. Replace “Trailer temperature and humidity…” for “Air temperature and humidity within the trailers...”.

Line 168. The change suggested was made. 

L175. Replace “Trailer temperature…” for “Air temperature…”.

Line 179. The suggested change was made. 

L199. Again, you are using a complex word, “attitude”, when carrying out behavioural studies. Think about and consider replacing for alertness or other word less complicate.

Thank you for your comment. The authors understand that the word ‘attitude’ may be complex, however we would like to keep this word because it is the word used in the scoring system described by Dewell and therefore we must use the terminology previously defined. In addition, this term was used in the two previous transport rest stop studies published in PLOS ONE and we would like to keep materials and methods as consistent as possible between the present and previous 2 studies. 

L246. Sound strange to include “weight” as a physiological trait

Materials and methods is divided into behavioural and physiological parameters. We believe it fits better under physiology than behaviour, therefore we have placed it in this section. 

336-351. Very hard to follow.

Line 344-360. We have made some changes to try and make the paragraph easier to understand. 

Results and discussion.

L356-358. Confusing, clarify.

Lines 365-369. Information was added to clarify. 

L358-360. Remove

We would prefer not to remove this sentence as it provides clarity if readers would wonder if differences could be due to animals not being equally distributed between compartments. 

L365-367. Triple interactions are usually hard to explain.

Agreed.

---

## [Decision Letter · Decision Letter 1]

21 Nov 2022

PONE-D-22-13599R1Effect of rest, post-rest transport duration, and conditioning on performance, behavioural, and physiological welfare indicators of beef calvesPLOS ONE

Dear Dr. Meléndez

Thank you for submitting your manuscript to PLOS ONE. After careful consideration, we feel that it has merit but does not fully meet PLOS ONE’s publication criteria as it currently stands. Therefore, we invite you to submit a revised version of the manuscript that addresses the points raised during the review process.

We look forward to receiving your revised manuscript.

Kind regards,

Marcio Duarte, PhD

Academic Editor

PLOS ONE

Journal Requirements:

Reviewers' comments:

Reviewer's Responses to Questions

**Comments to the Author**

1. If the authors have adequately addressed your comments raised in a previous round of review and you feel that this manuscript is now acceptable for publication, you may indicate that here to bypass the “Comments to the Author” section, enter your conflict of interest statement in the “Confidential to Editor” section, and submit your "Accept" recommendation.

Reviewer #3: All comments have been addressed

2. Is the manuscript technically sound, and do the data support the conclusions?

Reviewer #3: Yes

3. Has the statistical analysis been performed appropriately and rigorously? 

Reviewer #3: Yes

4. Have the authors made all data underlying the findings in their manuscript fully available?

Reviewer #3: Yes

5. Is the manuscript presented in an intelligible fashion and written in standard English?

Reviewer #3: Yes

6. Review Comments to the Author

Reviewer #3: PONE-D-22-13599-R1

Lines 177-178: inconsistencies in the description of the THI stress categories. Be consistent in terminology.

179-180: if this table reports air temperature, then there needs to be a unit of measure provided. And if i’m interpreting the table correctly, the only information that is presented is the THI. Revise title to be “Temperature Humidity Index (THI) within trailers….”

Lines 184: how was body temperature measured? Rectal temperature? When was this taken and how?

I see that the description of the handling/bw/temp evaluation is in lines 255-259. The metrics that were collected using this equipment are mentioned prior to introduction of the equipment. Consider reorganizing.

Line 446: apologies, but i’ve not seen results reported like this before. What is the 1=25.9 in reference to?

There are a lot of acronyms and abbreviations in this paper. The study design was complex, and thus the reporting will also be complex. This is to be expected. However, to facilitate easy readership, I recommend that the authors remind the reader periodically what the different acronyms represent. I find myself having to go back in the text regularly to remember what treatment combination I’m reading about. Can you help?

What are the little black dots on the figures? Are they supposed to be indicating statistical significance? If so, then that needs to be included in the figure title. In general, the figure titles need to be more comprehensive in their description.

Cortisol: what role do the author’s believe adrenal fatigue played in the outcome of their results? Perhaps the N calves didn’t have the internal resources to maintain cortisol production even though they are experiencing a stressful event?

7. PLOS authors have the option to publish the peer review history of their article (what does this mean?). If published, this will include your full peer review and any attached files.

Reviewer #3: **Yes: **Courtney Daigle

---

## [Author Response · Author response to Decision Letter 1]

21 Nov 2022

Lethbridge, November 21th, 2022

Dear Dr. Marcio Duarte,

Thank you very much for your message of November 21th, 2022, regarding our manuscript PONE-D-22-13599 “Effect of rest, post-rest transport duration, and conditioning on performance, behavioural, and physiological welfare indicators of beef calves”. We are grateful for the constructive comments of the reviewer. We hereby would like to submit the revised version of our manuscript for publication in PLoS One. 

Please find our detailed response below, with our responses to the reviewers’ comments in blue text. Changes in the manuscript have been highlighted in red.

The funders had no role in study design, data collection and analysis, decision to publish, or preparation of the manuscript

Yours sincerely,

Daniela Meléndez

Comments to the Author

1. If the authors have adequately addressed your comments raised in a previous round of review and you feel that this manuscript is now acceptable for publication, you may indicate that here to bypass the “Comments to the Author” section, enter your conflict of interest statement in the “Confidential to Editor” section, and submit your "Accept" recommendation.

Reviewer #3: All comments have been addressed-Thank you very much for reviewing this article. We are aware that there is a shortage of reviewers so we really appreciate that you took the time to review our paper. 

2. Is the manuscript technically sound, and do the data support the conclusions?

Reviewer #3: Yes

3. Has the statistical analysis been performed appropriately and rigorously? 

Reviewer #3: Yes

4. Have the authors made all data underlying the findings in their manuscript fully available?

Reviewer #3: Yes

5. Is the manuscript presented in an intelligible fashion and written in standard English?

Reviewer #3: Yes

Lines 177-178: inconsistencies in the description of the THI stress categories. Be consistent in terminology. Thank you for your comment. The terminology has been changed as suggested. Line 176. 

179-180: if this table reports air temperature, then there needs to be a unit of measure provided. And if i’m interpreting the table correctly, the only information that is presented is the THI. Revise title to be “Temperature Humidity Index (THI) within trailers….”

Thank you for your comment. That is correct. Temperature and humidity index is the only data presented in the table therefore no unit of measure is reported. 

Lines 184: how was body temperature measured? Rectal temperature? When was this taken and how?

I see that the description of the handling/bw/temp evaluation is in lines 255-259. The metrics that were collected using this equipment are mentioned prior to introduction of the equipment. Consider reorganizing.

Thank you for your comment. To address this we have move the ‘sample collection’ section after ‘morbidity and mortality’. Line 304-312. 

Line 446: apologies, but i’ve not seen results reported like this before. What is the 1=25.9 in reference to?

Thank you for your comments. We are reporting the upper and lower limits as described in lines 332-333. We understand that this is confusing as the letter ‘l’ and the number ‘1’ look very similar in times new roman font as you can see here: 1 l.

There are a lot of acronyms and abbreviations in this paper. The study design was complex, and thus the reporting will also be complex. This is to be expected. However, to facilitate easy readership, I recommend that the authors remind the reader periodically what the different acronyms represent. I find myself having to go back in the text regularly to remember what treatment combination I’m reading about. Can you help?

Thank you for your comment. We have added a treatment group and a sampling point keys at the beginning of the behaviour, performance, and physiology results sections to aid the reader with the acronyms. Lines 350-351, 508-509, 530-531, 649-652. 

What are the little black dots on the figures? Are they supposed to be indicating statistical significance? If so, then that needs to be included in the figure title. In general, the figure titles need to be more comprehensive in their description.

Thank you for your comment. Yes, the black stars that look like black dots represent significant differences p < 0.05. We have included the suggested information in the figure titles. 

Cortisol: what role do the author’s believe adrenal fatigue played in the outcome of their results? Perhaps the N calves didn’t have the internal resources to maintain cortisol production even though they are experiencing a stressful event?

This is a great point that we did not think about. We have added this information to the text. Line 673-674.

---

## [Editor Report · Decision Letter 2]

23 Nov 2022

Effect of rest, post-rest transport duration, and conditioning on performance, behavioural, and physiological welfare indicators of beef calves

PONE-D-22-13599R2

Dear Dr. Meléndez,

We’re pleased to inform you that your manuscript has been judged scientifically suitable for publication and will be formally accepted for publication once it meets all outstanding technical requirements.

Kind regards,

Marcio Duarte, PhD

Academic Editor

PLOS ONE

Additional Editor Comments (optional):

All comments were addressed. The manuscript is now ready to be accepted for publication.
---

## [Editor Report · Acceptance letter]

25 Nov 2022

PONE-D-22-13599R2 

Effect of rest, post-rest transport duration, and conditioning on performance, behavioural, and physiological welfare indicators of beef calves 

Dear Dr. Meléndez:

I'm pleased to inform you that your manuscript has been deemed suitable for publication in PLOS ONE. Congratulations! Your manuscript is now with our production department. 

Kind regards, 

on behalf of

Dr. Marcio Duarte 

Academic Editor

PLOS ONE